



# Technical note: Interpretation of field observations of point-source methane plume using observation-driven large-eddy simulations

Anja Ražnjević[1], Chiel van Heerwaarden[1], Bart van Stratum[1], Arjan Hensen[2], Ilona Velzeboer[2], Pim van den Bulk[2], and Maarten Krol[1,3]

[1]Meteorology and Air Quality Group, Wageningen University, Wageningen, The Netherlands
[2]Department of Environmental Modelling, Sensing & Analysis, Organisation for Applied Scientific Research, the Netherlands
[3]Institute for Marine and Atmospheric Research Utrecht, Utrecht University, the Netherlands

**Correspondence:** Anja Ražnjević (anja.raznjevic@wur.nl)

**Abstract.** This study demonstrates the ability of large-eddy simulation (LES) forced by a large-scale model to reproduce plume dispersion in an actual field campaign. Our aim is to bring together field observations taken under non-ideal conditions and LES to show that this combination can help to derive point source strengths from sparse observations. We prepared a one-day case study based on data collected near an oil well during the ROMEO campaign (ROmanian Methane Emissions from Oil and gas) that took place in October 2019. We set up our LES using boundary conditions derived from the meteorological reanalysis ERA5 and released a point source in line with the configuration in the field. The weather conditions produced by the LES show close agreement with field observations, although the observed wind field showed complex features due to the absence of synoptic forcing. In order to align the plume direction with field observations, we created a second simulation experiment with manipulated wind fields. The estimated source strengths using the LES plume agrees well with the emitted artificial tracer gas plume, indicating the suitability of LES to infer source strengths from observations under complex conditions. To further harvest the added value of LES, higher order statistical moments of the simulated plume were analysed. Here, we found good agreement with plumes from previous LES and laboratory experiments in channel flows. We derived a length scale of plume mixing from the boundary layer height, the mean wind speed and convective velocity scale. It was demonstrated that this length scale represents the distance from the source at which the predominant plume behaviour transfers from meandering dispersion to relative dispersion.

*Copyright statement.* TEXT

## 1 Introduction

The reduction of greenhouse gases (GHG) emissions is of the highest importance in mitigation of climate change. Methane (CH$_4$) is one of the most potent GHGs, but due to its relatively short lifetime in the atmosphere, reduction of CH$_4$ emissions can have more immediate positive effects on the mitigation of climate change effects (e.g. Baker et al. (2015); Zickfield et al. (2017); Caulton et al. (2018)). Methane has large variety of sources that differ in origin (anthropogenic or natural) and size (





e.g. point-like, diffuse, line). An overview of different source types and their contribution to global methane budget is given by
Saunois et al. (2016).

In order to help constrain methane emissions, the Methane goes Mobile – Measurements and Modelling (MEMO$^2$) project
started in 2017. The goal of the project was to improve $CH_4$ emission factors in inventories on European scale by combining
extensive measurement campaigns of different sources of $CH_4$ with modelling techniques across different scales. The MEMO$^2$
consortium participated in a campaign in which methane emissions from Romanian oil and gas industry (ROMEO) were sam-
pled. The campaign took place during October 2019. Sources of methane were measured on basin and well scales employing
various measurement techniques.

With methane often being released from small but strong sources in a turbulent atmosphere, the observation of plumes is chal-
lenging. A large variety of measurement techniques have been developed for identification and quantification of GHG sources,
ranging from satellite based observations (e.g. Bergamaschi et al. (2007); Jacob et al. (2016)) and basin-scale measurements
using aircraft (Conley et al., 2017) to local source measurement techniques. These local techniques include, among others,
instruments placed on unmanned aerial vehicles (UAVs) (e.g. Andersen et al. (2018); Shah et al. (2019)), instruments placed
on ground vehicles (e.g. Hensen et al. (2006); Baillie et al. (2019)) and point measurements from sensors mounted on towers
(Röckmann et al., 2016). Each of these techniques has its own strengths, either being highly accurate in time or covering large
spatial areas, but neither does both. Dispersion models provide insight into the behavior of plumes and can help with the data
interpretation and planning of measurement strategies. These models vary greatly in their complexity and underlying assump-
tions. Most commonly, Gaussian plume models are combined with observations to quantify sources (Caulton et al., 2018; Edie
et al., 2020; Rybchuk et al., 2020). These simple models are fast and easy to use but come with restrictive assumptions (e.g.
stationarity of the plume and the mean wind) that make their application challenging under the strongly transient conditions
that often characterize the local atmospheric boundary. With the development in computer power in the past decades, high res-
olution models that are able to simultaneously resolve the turbulent velocity field and describe the transport of emitted tracers,
large eddy simulations (LES), have been increasingly used for plume studies (Cassiani et al., 2020). LES explicitly resolve the
largest eddies, which carry most of the energy, and parameterize the smallest scales using subgrid-scale models (e.g.Deardorff
(1973); Pope (2000)). LES have been utilized in many dispersion studies, mostly focusing on idealized channel flows in various
stability regimes (e.g. Dosio & de Arellano (2006); Boppana et al. (2012); Ardeshiri et al. (2020)). LES have been successfully
validated (e.g. Dosio & de Arellano (2006); Ardeshiri et al. (2020)) against a considerable amount of extensive laboratory dis-
persion studies. These dispersion studies include channel flows in either water or air (e.g. Fackrell & Robins (1982a, b); Gailis
et al. (2007); Nironi et al. (2015)). One of the main advantages of LES in dispersion studies is that it provides a high temporal
and spatial resolution of the 3D plume. This enables detailed analysis of the plume statistics, something which is difficult to do
only from observations. Furthermore, LES can be used as a laboratory for optimizing measurement strategies. Despite the vast
amount of idealized studies, the performance of LES has not been validated against many experimental field studies. (Steinfeld
et al., 2008; Ardeshiri et al., 2020; Rybchuk et al., 2020). For instance, the Prairie Grass experiment (Barad, 1958) still serves
as a common reference for LES studies. More recently, Caulton et al. (2018) evaluated the performance of the Gaussian plume
model against measurements in a neutral atmosphere and LES while Rybchuk et al. (2020) evaluated WRF LES with Prairie





Grass experiment for convective conditions.

In this study, we aim to bring together actual field observations under less than ideal conditions with LES. The ROMEO campaign focuses on sampling methane from spatially distributed sources covering a large area using the mobile measurement techniques. While this approach is very useful in detecting unknown sources, measurements of individual, isolated plumes are often sparse. This is due to the measurement techniques employed. For example, plume transects using cars only provide observations at the surface in one dimension. Moreover, this observation strategy is limited by the conditions in the field (accessibility of the source, the amount of adjacent roads on which measurements can be taken, road conditions etc.). Here, we aim to demonstrate how LES can help in interpreting sparse observations in the field both from a viewpoint of validation and interpretation. We will present an LES dispersion study of a methane plume measured during the ROMEO campaign. The LES study is set-up combining local meteorological observations with ECMWF ERA5 (Hersbach et al., 2020). We will compare the measured plume with the LES simulated plume in order to gain insight in the information gathered through the measurement process, and to evaluate the performance of LES. Finally, we will use LES to study the structure of the simulated plume in convective conditions and its behavior by analyzing the higher order statistical moments. We will build on the idealized studies Dosio & de Arellano (2006); Cassiani et al. (2020) and apply the statistical analyses to simulations that represent realistic field conditions.

The structure of this paper is as follows: In Section 2 we introduce the location where the measurements took place and look at the meteorological conditions relevant for the plume dispersion. Following this, in Section 2.2 we present available data from the campaign as well as the methods and instruments employed in the field. In Section 3 we present the numerical model used to perform the LES, the simulation set-up and forcing used to reproduce the meteorological conditions. In this section we will also outline the statistical methods used to inspect the simulated plume behavior. In Section 4 we evaluate the LES plume with observation and discuss the characteristics of the both plumes. This is followed by a more in-depth analysis of the simulated plume at various distances from the source. Finally, in Section 5 we evaluate the usability of LES for dispersion studies under realistic meteorological conditions.

## 2 Case description

### 2.1 Site description and meteorological conditions

The case study presented here is based on measurements performed by a team from the Netherlands Organisation for Applied Scientific Research (TNO) during the ROMEO campaign in Romania. The measurements were taken in the Parhova County. The county is characterized by plains in the South and the Carpathian mountains in the North. The actual site is located in the central part of the county, where the two distinct landscapes meet. The measurements were performed on the road downwind from an oil well over the course of 3 h. The length of the road segment on which the plume was measured was 150 m, while





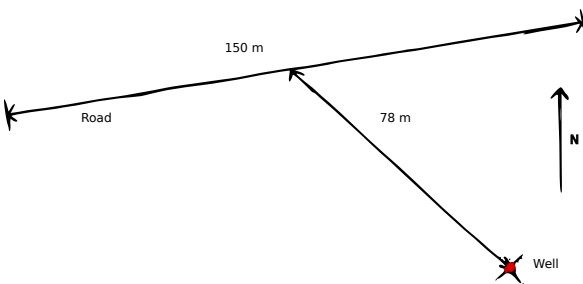

**Figure 1.** Sketch of the oil well location and the adjacent road the measurements were preformed on.

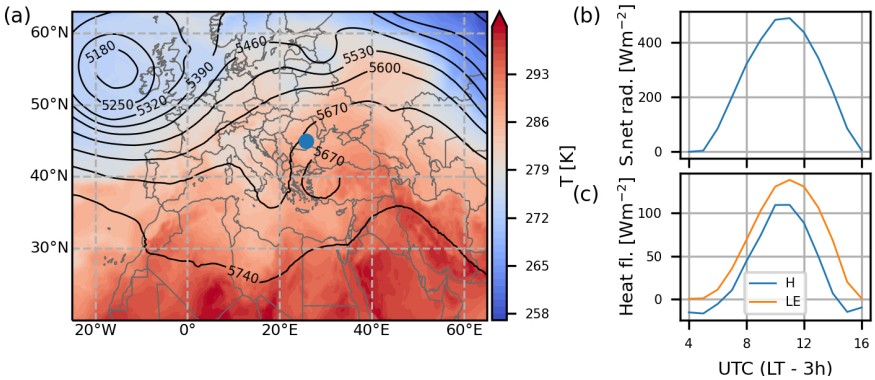

**Figure 2.** Meteorological situation over Europe on 17th October 2019. (a) Geopotential height (m) at the 500 hPa pressure level and temperature at 850 hPa at 12 UTC. Location of the studied region is indicated on the map by a blue circle. (b) Surface net solar radiation, (c) hourly values of sensible an latent heat at the location of the measurements.

the mean distance from the road to the well was 78 m. Note here that the Figure 1 shows the sketch of the well location and the adjacent road as positioned in the North - East coordinate system and not in the relation to the mean wind.

    The measurements were performed in the afternoon (14.30 - 17.30 LT (UTC + 3h)) on 17 October 2019. The weather at the location was characterized by very low winds and no cloud cover, as confirmed by the participants of the campaign. To analyse

the overall synoptic situation over Europe and Romania in particular, we used the geopotential heigh and temperature chart obtained from ERA5 (Hersbach et al., 2020). Figure 2 a shows a low pressure system situated in the Atlantic ocean close to the Irish coast. A high pressure system was located over Russia with the highest gradients between the two systems extending over the Baltic region. The weather over Romania was characterized by very low gradients in both temperature and pressure, resulting in low wind speeds and advection during the campaign. Hence, we expect that the conditions were strongly influenced

by local convection.



The Figure 2 shows the surface net solar radiation and the sensible and latent heat fluxes in the region (panels (b) and (c) respectively) retrieved from the ERA5 data. The surface heat fluxes are comparable indicating dry conditions (Fig. 2 b), and the solar radiation indicates no clouds were present. Furthermore, it was inferred from the ERA5 data that the temperature maximum at the surface was 22°C and the boundary layer (BL) depth reached 700 m at 12 UTC (not shown). The ERA5

height wind profiles show wind turning with height. Maxima in wind speeds in opposite direction from those at the ground were seen at approximately 1000 m height, or just above the BL top. For the duration of the measurements, the mean horizontal wind at the surface showed variation of 10° (not shown).

As will be presented later in Figure 3, we find that, for the duration of the measurements, the temperature at 1000 hPa level remained almost constant, and showed almost linear decrease with height. Similarly, the specific humidity showed peak values

at the surface and constant values in the mixing layer above it. Above the BL, specific humidity decreased with height. The near constant in time height profiles of specific humidity and temperature confirm that the large scale advection was small during the campaign. As a result, the time evolution of temperature and humidity was determined by local processes. This leads to a complex pattern in vertical baroclinicity, which is a challenge for both collecting experimental data and simulation studies.

## 2.2   Measurement instruments and available data

The measurement device used was a dual laser trace gas monitor based on Tunable Infrared Laser Direct Absorption Spectroscopy (TILDAS; Aerodyne Research Inc., Billerica, US) that measures methane and ethane ($CH_4$ and $C_2H_6$) simultaneously. The ethane data is used to discriminate methane plumes originating from fossil fuel related sources (which contain ethane) and agricultural or biomass degradation methane emissions. The instrument also measures $H_2O$, $CO_2$, CO and $N_2O$. The concen-

tration levels for these components are measured at sub-ppb resolution with 1 measurement per second (1 Hz). Concentration levels were calibrated versus working standards for $CH_4$ (B20 flasks with compressed air at 2800 and 5000 ppb) that are linked to the Integrated Carbon Observation System (ICOS) & the National Oceanic and Atmospheric Administration (NOAA) standards used at the Cabauw tall tower in the Netherlands. The instrument was placed into a vehicle that drove along the closest road and its position was logged at 1 Hz with a GPS system. The inlet of the measurement device was placed at the top of the

vehicle at a height of 3 m. A delay-time correction is applied to the data to compensate for the  1.5 second delay between the logged GPS location and actual measurement in the TILDAS instrument.

The wind speeds ($u$, $v$, $w$) were measured from a battery-operated Gill R3 sonic anemometer placed close to the source and at 1.8 m above ground level. The sonic data was stored at 20 Hz, and 1 Hz values were transmitted with a wireless link to the central computer in the van. For this analysis, 1 min averages of wind speeds were used.

Due to the road conditions the vehicle speed varied from transect to transect. Therefore, the measured plumes have a different number of measured points and the exact distribution of these points over the measurement transect differs. To obtain a uniform dataset, two end points were selected that encompass all plumes. Values of the vehicle location, $CH_4$ and $N_2O$ data were linearly interpolated on a 250 point grid.



# 3 Numerical methods

Large-eddy simulations were performed using the MicroHH model, which is an open-source computational fluid dynamics code (van Heerwaarden et al., 2017). The code solves conservation equations of energy, momentum, and mass under the anelastic approximation. The transport of passive scalars is solved with the advection-diffusion equation. The second-order Smagorinsky model is used for the subgrid parametrization of the velocity components.

Time integration is performed using third-order accurate Runge-Kutta scheme and the spatial domain is discretized on a staggered Arakawa-C grid.

The advection term for dispersing scalar in the model is solved using a second-order energy conserving scheme. For atmospheric transport, positivity in numerical schemes plays a crucial role. By imposing positivity using a flux limiter, over- and under-shoots are avoided in areas with strong concentration gradients (Hundsdorfer et al., 1995).

A periodic boundary condition was imposed for the momentum and thermodynamic variables on the lateral boundaries of the domain. The lower boundary had no-slip ($u = v = 0$) and no penetration ($w = 0$) boundary conditions, while the upper boundary had free-slip boundary conditions, with tangential components of velocity being zero ($\frac{\partial u}{\partial z} = \frac{\partial v}{\partial z} = 0$). The inflow and outflow boundary conditions for the scalar representing $CH_4$ were imposed at the lateral boundaries of the domain to prevent the plume from re-entering. The boundaries were set using Neumann (right and lower boundaries) and Dirichlet (left and lower boundaries) boundary conditions which were used to interpolate values of scalars in two ghost cells outside of the domain.

In order to achieve LES that corresponds to the field conditions large scale forcings of relevant variables is imposed by coupling the LES simulations with the ERA5 data (Hersbach et al., 2020). The geostrophic wind and large scale advection terms are interpolated from the ERA5 data, while the nudging of the simulation is applied on a relevant timescale to prevent the simulation from drifting from the large-scale mean profiles while smaller scale turbulence can still develop independently. The coupling is based on Schalkwijk et al. (2015).

## 3.1 Simulation set-up

The LES was performed in a three dimensional domain of $4.8 \times 4.8 \times 3.085$ km (x, y and z direction respectively). The domain was discretized on a $960 \times 960 \times 480$ (x, y, z) grid. This results in uniform horizontal resolution of 5 m, while the vertical direction is resolved on a stretched grid with 2 m resolution in the first 1 km of the domain and 50 m at the top. The resolution was chosen with the computational feasibility in mind such that the grid was dense enough for the dispersion to be well represented, but still have the domain large enough to capture the meteorological effects relevant for this study.

A constant source of passive scalar was added as a two-dimensional Gaussian placed on the bottom of the domain. The Gaussian had the $1\sigma_i$ (i = x, y) value equal to size of one grid box, therefore 97% of the source was spread on $4^2$ grid. It has been shown in laboratory studies of plume dispersion that the ratio of the source size and the size of larger scale eddies has significant impact on plume statistics (Fackrell & Robins, 1982a, b; Nironi et al., 2015). To circumvent this issue, the size of the source should be larger than the size of one grid box. Recently, Ardeshiri et al. (2020) investigated the influence of resolution on the flow and plume statistics in LES. They have shown that the scalar variances converge for the sources resolved by at least $4^3$





**Table 1.** The specifics of the two performed simulations. First row shows set-up of the simulation with highly turning mean wind and the second the set-up of the simulation with the mean along the x axis.

| Simulation | Domain size (km) | Resolution | Source position (m) | Geostrophic wind | Tendencies: $u$ and $v$ | Tendencies: $\theta_l$ and $q_t$ |
|---|---|---|---|---|---|---|
| Realistic | $4.8 \times 4.8 \times 3.085$ | $960 \times 960 \times 480$ | (3600, 3600, 0) | On | On | On |
| Idealized | $4.8 \times 4.8 \times 3.085$ | $960 \times 960 \times 480$ | (480, 2400, 0) | Off | Off | On |

grid nodes. We have placed the source in the top right corner of the domain at the position (3600, 3600) m, where the domain origin is defined at the lower left corner. The scalar was emitted with constant flux of 1 g s$^{-1}$.

In order to reproduce meteorological conditions in the fields encountered on the measurement day, the simulation was nudged
according to height profiles of horizontal wind speed, temperature and specific humidity obtained from the ERA5 data. Large scale forcing was imposed through geostrophic wind with the Coriolis parameter for this latitude being $f_C = 1.0305 \cdot 10^{-4}$ s$^{-1}$. Furthermore, roughness lengths of $z_{0h} = 0.001$ m and $z_{0m} = 0.05$ m were imposed on the lower boundary for scalar and momentum respectively.

The simulation was run for 7.5 hrs in total. A spin-up time was imposed to have the boundary layer fully developed and resembling field conditions as closely as possible. First, the fields from ERA5 were initiated for 7 AM UTC and the simulation was started with an integration time step of 6 s. The simulation was run for 7 hrs with vertical profiles being nudged towards ERA5 profiles every hour. At 11 UTC, the source was activated in the simulation. The instantaneous plume concentrations $c$, wind components $u$, $v$ and $w$ and liquid potential temperature $\theta_l$ were recorded on various two-dimensional cross-sections of
the domain.

The mean wind in this simulation shows fluctuating behavior, which influences the direction the plume dispersion. Here, we assume that the local wind that influenced the dispersion was governed by local influences that are not captured in ERA5. To be able to still compare the plume at different simulation times, the mean wind speed between observations and simulation
should be aligned. Therefore, we performed another simulation in which the mean wind was directed along the x axis, while keeping all the other specifics, apart from the source location, identical. These two simulations will be further on referred to as *realistic* and *idealized* for the first and second simulation, respectively. The overview of the specifics of the two simulations are given in Table 1. Note that in the idealized simulation, the nudging profiles for potential temperature and specific humidity still originate from the ERA5 dataset. For the wind, however, we set the height profile of the $v$ component in the nudging
profiles to zero and set the $u$ profile to a constant value of 3 m s$^{-1}$. This wind speed was chosen through manual tuning to get a good match with the measured wind speed at 2 m height. In this way the wind direction was kept constant without loosing the general characteristics of the realistically simulated boundary layer.





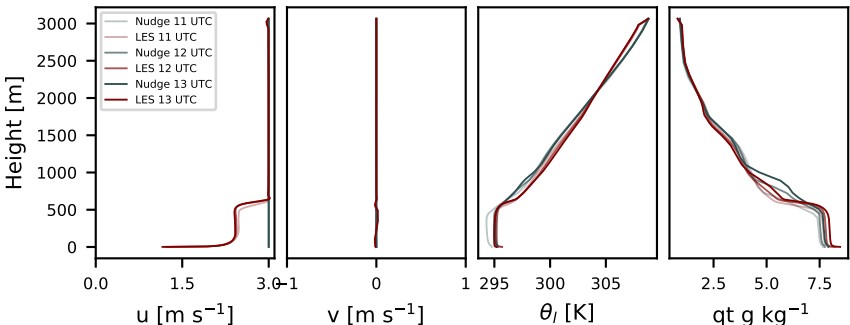

**Figure 3.** The nudging height profiles and the hourly mean height profiles of variables from the simulation with with centered mean horizontal wind direction.

### 3.2 Estimation of the unknown emission rate

Source quantification from one-dimensional transect measurements is often performed using a mass balance approach. This method compares the total line-integrated flux of the time averaged plume from an unknown source with the flux from a known source under the same atmospheric conditions and at the same downwind distance from the source. This method has been used in conjunction with either a tracer release, or - if no tracer is co-emitted - with simple plume transport models such as the Gaussian plume model (e.g. Caulton et al. (2018)). The equation used for this approach reads:

$$Q_{estim} = \frac{\sum_{y} C_{meas}\overline{u_{meas}}}{\sum_{y} C_{ref}\overline{u_{ref}}} \times Q_{ref}. \tag{1}$$

Here the $Q_{estim}$ is the emission rate of the unknown source in g s$^{-1}$, $C$ g kg$^{-1}$ and $\overline{u}$ m s$^{-1}$ denote the time-averaged measurements and the mean wind speed, respectively. Subscripts $ref$ and $meas$ refer to the reference tracer with known source $Q_{ref}$ and measured tracer, respectively. Note that the reference plume can be either measured of inferred from a model.

### 3.3 Statistical properties of the modeled plume

Finally, we present a short overview of the statistical moments calculated for the simulated plumes that will be discussed in Sections 4.2.1 – 4.2.3. Higher order statistics can provide further insight into the behavior of the measured plume but are often unattainable from the measurements due to insufficient spatial and temporal resolution. Figure 4 shows a scheme of an idealized plume emitted from a ground point source. Let $z$ be the vertical position of a particle in an instantaneous plume at a distance x from the source. This plume is characterized by its centerline position $z_m$ defined as its center of mass:

$$z_m(x,t) = \frac{\int c(x,y,z,t)\, z\, dz\, dy}{\int c(x,y,z,t)\, dz\, dy}. \tag{2}$$

An ensemble of such instantaneous plumes will have its own centerline position $\overline{z_m}$ defined as the mean over all the realizations. Now the fluctuations of the instantaneous plume around these mean positions can be defined. The absolute fluctuation $z'$





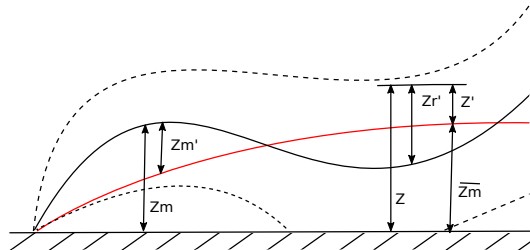

**Figure 4.** Scheme of idealized plume dispersing from a ground point-source. Shown here are the mean position of the instantaneous plume $z_m$ and the centerline position and the total mean of centerline positions $\overline{z_m}$ (red line). Also shown are the displacements of particles in the instantaneous plume from its centerline position $z_r$ and from the total mean centerline position $z'$ as well as the displacement of the instantaneous centerline from the total mean centerline $z'_m$.

is the displacement of the particle in the plume from the mean centerline position $\overline{z_m}$, relative fluctuation $z'_r$ is the displacement from the instantaneous plume centerline $z_m$ and the fluctuation of the instantaneous plume centerline $z'_m$ is the displacement from the mean position $\overline{z_m}$. These can be written as

$$z' = z - \overline{z_m}, \quad z'_r = z - z_m, \quad z'_m = z_m - \overline{z_m}. \tag{3}$$

The mean plume positions and the displacements in the y direction can be defined in a similar manner.

Following the definition of Nieuwstadt (1992), the absolute plume dispersion, or the second-order moment, in the vertical direction is written as

$$\sigma_{za}^2(x,t) = \frac{\int c(x,y,z,t)\, z'^2 \, dy\, dz}{\int c(x,y,z,t)\, dy\, dz}. \tag{4}$$

The absolute plume dispersion can be decomposed to its meandering and relative contributions, or dispersion due to movement of the plume centerline, or meandering, and diffusion of particles from the plume centerline. Therefore, it holds

$$\sigma_{za}^2 = \sigma_{zm}^2 + \sigma_{zr}^2 \tag{5}$$

where

$$\sigma_{zm}^2(x,t) = \frac{\int c(x,y,z,t)\, z'^2_m \, dy\, dz}{\int c(x,y,z,t)\, dy\, dz}, \quad \sigma_{zr}^2(x,t) = \frac{\int c(x,y,z,t)\, z_r^2 \, dy\, dz}{\int c(x,y,z,t)\, dy\, dz}. \tag{6}$$

The third-order moment is therefore

$$\overline{z_a}^3(x,t) = \frac{\int c(x,y,z,t)\, z'^3 \, dy\, dz}{\int c(x,y,z,t)\, dy\, dz}, \quad \overline{z_r}^3(x,t) = \frac{\int c(x,y,z,t)\, z_r^3 \, dy\, dz}{\int c(x,y,z,t)\, dy\, dz}, \quad \overline{z_m}^3(x,t) = \frac{\int c(x,y,z,t)\, z'^3_m \, dy\, dz}{\int c(x,y,z,t)\, dy\, dz}. \tag{7}$$

Similar expressions hold for moments in the y direction. Lastly, we define skewness here as $S_i = \frac{\overline{i^3}}{\sigma_i^3}$, where i = (y, z).





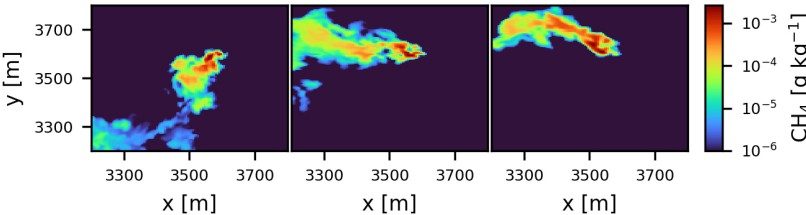

**Figure 5.** Snapshots of instantaneous plumes at 3 m above the surface. Plume at (left) 11:30:00 UTC, (middle) 12:11:30UTC, (right) 13:00:00 UTC.

## 4 Results

### 4.1 Validation of modeled meteorological conditions with available data

Figure 5 shows multiple instantaneous xy cross-sections of the simulated plume from the *realistic* simulation. It can be seen that the variation of the plume direction is pronounced throughout the simulation. This behavior is caused by very low mean wind speeds that change direction frequently in the simulated turbulent flow field. As was demonstrated in Section 2, the large-scale wind and temperature fields showed no pronounced gradients above the area on the simulated day. Therefore, local effects likely governed the flow that was measured on the site. Since ERA5 does not resolve these local effects, the discrepancies

between modeled and measured wind are to be expected. To correct for this, the *idealized* simulation was set up.

Simulated and nudging profiles for the *idealized* run are compared in Fig. 3. Note that we replaced the ERA5 wind profiles. The simulated profiles were obtained as spatial averages over the whole domain that were time-averaged over one hour. Throughout the run, $\theta_l$ and $q_t$ show very good agreement with the ERA5 profiles. The profiles show hardly any variability over

time, and are constant with height in the lower $\approx 700$ m. This indicates that the spin-up time of the run was sufficiently long for the development of a well-mixed boundary layer. The $u$ wind profile in the *idealized* run shows virtually no variation. A well-mixed layer above the surface is clearly visible, with stronger and constant winds above 700 m, which correspond to the nudging profiles, and a logarithmic decline towards the surface. The simulated $v$ profiles agree with the imposed zero nudging profiles.


Figure 6 a shows the measured $CH_4$ and $N_2O$ plumes. The plumes in the figure are shown with the background subtracted. Background subtraction follows the procedure described in Ruckstuhl et al. (2012). To help the interpretation of the plumes, $N_2O$ was released from a cylinder 20 cm above the well head with the constant rate of 0.59 g s$^{-1}$ for the duration of the measurements.

Them measured horizontal wind speed (Fig. 6 b) was low during the the whole day, varying from 0.8 m s$^{-1}$ to 2.8 m s$^{-1}$ (1-minute averages) during the measurements. A weak periodicity of approximately 55 min can be noticed in the wind speed





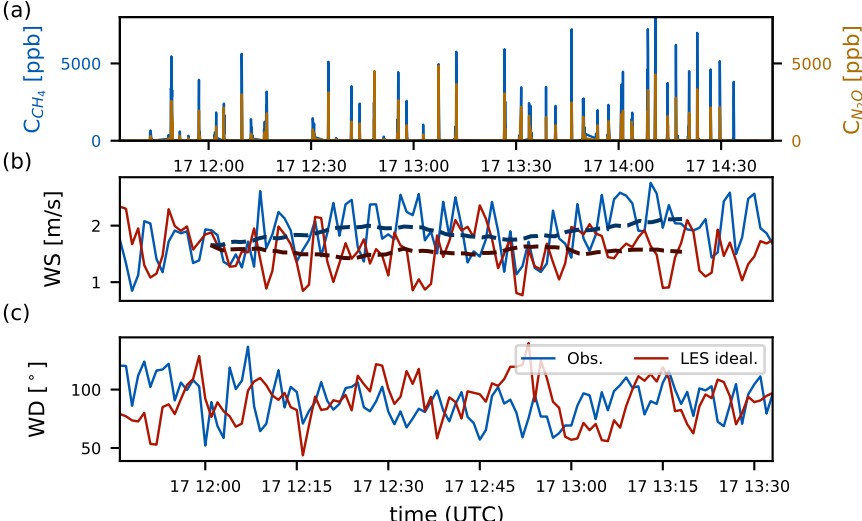

**Figure 6.** (a) $CH_4$ concentrations measured over the read adjacent to the emitting oil-well and $N_2O$ concentrations emitted from next to the well. Comparison of the observed and simulated (b) horizontal wind speed, (c) horizontal wind direction. The values are given as one minute averages of instantaneous wind. The dotted lines are rolling means of respective wind speeds, shown here for easier interpretation of the mean wind speed.

data. It is possible that it is caused by influences from the local orography, since the area is in vicinity of hills. The closest elevated area is about 5 km away towards W, and higher mountains are located approximately 10 km towards N and NW. To verify this claim, however, the wind data should be considerably longer. The periodicity, on the other hand, is not present in the
simulated wind (Fig. 6 b). This is caused by the imposed flat orography in the domain and the constant wind forcing through the nudging on the lateral boundaries. Nevertheless, the simulated and measured wind speeds show good agreement. This is also visible in the 1 min averages of the instantaneous wind direction (Fig. 6 c). Since the mean wind direction in the *idealized* simulation was set to be easterly, the wind was rotated to match the wind direction from the observations. It can be seen that the wind angle in the *idealized* simulation fluctuates comparable to the observations.

**4.2   Comparison of modeled and measured plume characteristics**

Time-averaged plumes from the measurements are given in the Fig. 7. The measured plumes were averaged over half hour increments, and are shown here together with the mean wind speed and direction for the corresponding time period. The mean horizontal wind direction did not change significantly during the measurement period, except for the first half hour, which deviates by $\approx 40°$. The mean wind speed for the whole period was small and did not exceed 2.4 m s$^{-1}$. Since the
measurements were collected on a public road, the number of averaged plumes varies from one half-hourly period to another. The number of plumes per half-hourly time period amounts to n = [3, 6, 8, 4, 9, 10] starting from 11.30 UTC until 14.30 UTC. On the right-hand panel of Fig. 7 the corresponding time-averaged simulated plumes are shown. Transects through the plume





were sampled according to the observations: at 3 m height and 80 m downwind from the source. Plume transects were taken

every 1 min, resulting in 150 transects for the entire simulation. After 14.00 UTC the surface flux of potential temperature

turns negative and that is when the simulation stops. For that reason, the set of time-averaged plumes from LES contains

one less element than the measurement set. The half-hour averages of the simulated plumes look smoother compared to the

corresponding measured plume averages. This is because the number of plumes averaged per half-hour increment is much

lower in the measurements compared to the simulation.

The angle over which the wind direction varies during the simulated period amounts to $18°$, with the exception of the last

half-hourly period period, in which the wind direction deviated from the mean by $\approx 50°$.

We used Eq. 1 to infer the unknown CH4$_4$ emission rate from the oil well using the LES plume as the reference. To this end,

we compare the time-averaged measured $CH_4$ plume from the oil-well (red line in Fig. 7 (b)), combined with the measured

mean horizontal wind speed ($\overline{u_{meas}}$ = 1.93 m s$^{-1}$), to the corresponding flux of the time-averaged LES plume (red line in Fig.

7 (c)), combined with the simulated wind speed ($\overline{u_{ref}}$ = 1.72 m s$^{-1}$). This leads to the estimated emission rate of $Q_{estim,CH_4}$

= 1.11 g s$^{-1}$. Using the same principle we estimate an $N_2O$ emission rate of $Q_{estim,N_2O}$ = 0.53 g s$^{-1}$ (true emissions was

$Q_{N_2O}$ = 0.59 g s$^{-1}$). To benchmark the performance of LES in this experiment, we also estimate the unknown $CH_4$ source

using the the $N_2O$ gas as reference, and obtain an emission rate of $Q_{estim,CH_4}$ = 1.23 g s$^{-1}$. The discrepancies between both

methods can have various reasons. Firstly, even though the mean wind in the LES is very close to the measured mean wind,

their magnitudes and variations are not identical, which can also contribute to the error. However, the most notable cause for

the estimation error might arise from the averaging time of the measurements, which is likely too short. It can be observed

from Fig. 7 that the time-averaged plumes of $CH_4$ and $N_2O$ are not Gaussian shaped, which indicates that turbulent eddies of

various sizes still influenced the time-averages. Nevertheless, we have shown that LES is a useful tool in source estimation in

real-atmosphere conditions, e.g. in cases for which the source location is inaccessible for a tracer release.

### 4.2.1    Absolute dispersion

In this section, we analyze the general behavior of the LES plume through its first three statistical moments i.e. the center of

mass, width of the plume and skewness. As shown in the previous section, the typical mobile plume measurements consist of a

relatively small number of 1D plume transects. While such measurements might be well suited for inferring the unknown source

strength, we aim here to exploit LES further. We will do this by linking the simulated plume to previous, more idealised, plume

dispersion studies. First we focus on the absolute motion of the plume with respect to the ground surface, and will statistically

analyse plume dispersion due to plume meandering and due to motions relative to the plume center of mass. In a later stage,

we will separate the two, as described in Section 3.3, and analyse the contribution of the two processes separately.

Figure 8 shows instantaneous and time averaged plumes. The plumes were integrated over depth (x-y transect), and width (x-z

transect). Clear differences in the structure of the plume can be observed between the instantaneous and time-averaged shape.

Firstly, the top-view on the time averaged plume (panel b) shows a clear Gaussian shape as expected, which deviates from the

instantaneous plume shown in panel (a). In the instantaneous plume, eddies of different sizes influence the plume throughout.

The integrated x-z transect of the plume shows more complex behavior. The solid line in the bottom panel of Fig. 8 shows the



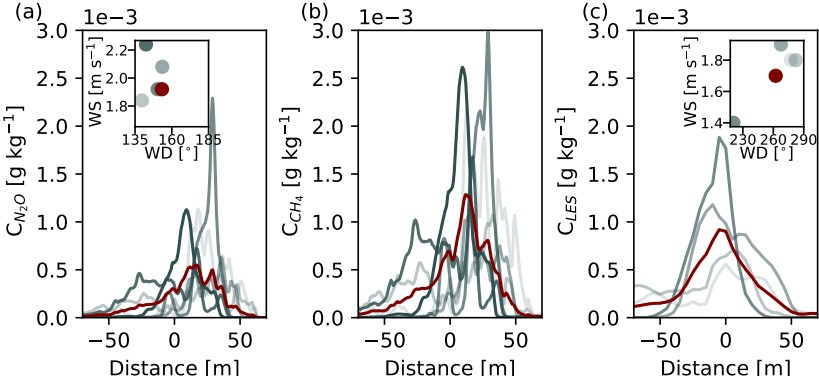

**Figure 7.** Averages of instantaneous plumes over periods of half hour from (a) $N_2O$, (b) $CH_4$ and (c) LES. LES transects were taken at 3 m height and 78 m downwind from the source. Plumes are shown with a color gradient corresponding to the half-hour increments i.e. lightest gray plume is the average of plumes measured over the time period 11.30 - 12.00 UTC, dark grey is the average over 14.00 - 14.30 UTC. The inset shows horizontal wind speed and direction for the corresponding half-hour averages in (a) measurements and (c) LES. Overplotted in red is the average of all the plumes, as well as the averages of horizontal wind speed and direction in the insets.

mean of all centerline plume positions as defined by Eq. 2. In contrast, the dotted line denotes the position of the maximum concentration of the integrated x-z plume. Firstly, it can be noticed that the positions of the maximum and the mean do not coincide close to the source (x < 2500 m), while at large distances the two lines tend to converge. Secondly, the position of the

maximum concentration is located at the ground level for x < 2000 m. For larger distances, the maximum concentration moves towards the top of the boundary layer and a local minimum is visible at the surface. Dosio & de Arellano (2006) performed a turbulent channel flow study with a similar set-up as we presented here. They presented their results as a function of the normalized distance $x_*$, defined as:

$$x_* = \frac{w_*}{h_{BL}} \frac{x}{\overline{u}}, \tag{8}$$

where $w_*$ is the convective velocity scale, $h_{BL}$ is the height of the boundary layer, $x$ is the distance from the source and $\overline{u}$ is the mean wind speed over the whole domain. Intuitively, this distance quantifies the number of overturns of the largest eddies (convective timescale $T_M = \frac{h_{BL}}{w_*}$) at distance $x$ from the source (advective timescale $T_A = \frac{x}{\overline{u}}$). Dosio & de Arellano (2006) reported similar behavior in the mean of their plume emitted from an elevated source. In that simulation the concentration maximum was first transported towards the surface and later lifted towards the boundary layer top. The corresponding local

minimum at the surface occurred at $x_* = 1.75$. In our simulation, with $w_* = 0.94$ m s$^{-1}$, $\overline{u} = 2.64$ m s$^{-1}$, and $h_{BL} = 564.64$ m, the minimum occurs at $x_* = 1.9$, which is in good agreement with the results of Dosio & de Arellano (2006). As mentioned previously, the position of the concentration maximum converges to the plume centerline position at larger distances from the source. This result, however, differs from the results reported by Dosio & de Arellano (2006). In their simulation, the plume never reaches a well-mixed state, which was in agreement with water tank experiments performed by Willis & Deardorff

(1978) who reported well-mixed plume only at very large distances from the source at $x_* = 6$. In comparison, in our simulation

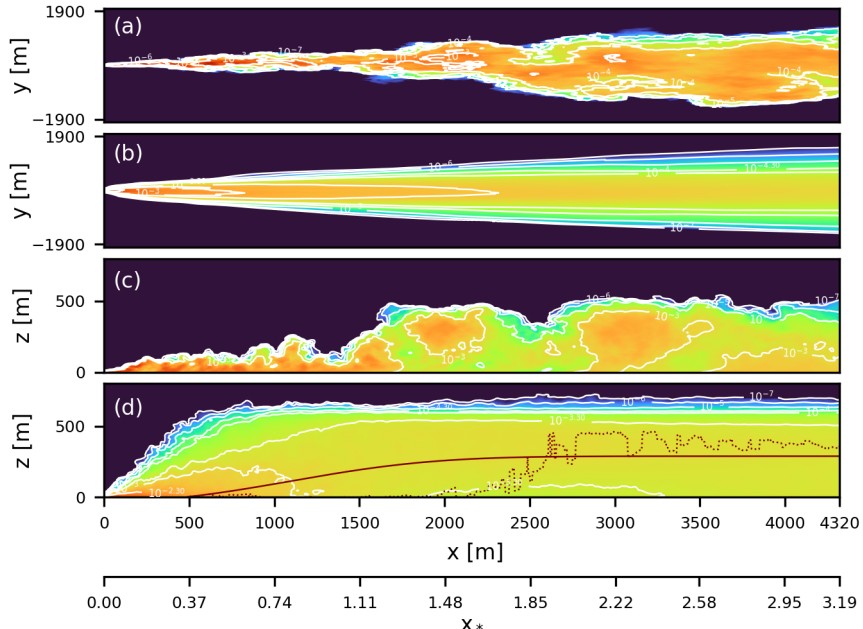

**Figure 8.** Top view on vertically integrated (a) instantaneous and (b) time averaged plume. Horizontally integrated (c) instantaneous and (d) time-averaged plume. On (d) are also shown mean plume centerline position (solid red line) and position of the maximum of concentrations (dotted red line). $x_*$ is the normalized distance from the source as defined in Eq. 8.

the maximum of concentration starts approaching the mean plume position at $x_* \approx 2.5$. The instantaneous plume we show here (Figure 8, panel (c)), has stayed close to the ground for $\approx 1500$ m from the source, after which it gets mixed in with larger sized eddies towards the top of the boundary layer. Additionally, puff-like structures with higher concentrations can be observed, which have been lifted from the ground and are carried to the top of the BL even at large distances from the source
(e.g. x = 3000 m).

Figure 9 shows the first three statistical moments of the plume. The top two panels show the plume centerline position in the y and z direction, downwind from the source. The centerlines of the instantaneous plume positions in y direction show large variability throughout the domain. However, the mean center of mass is constant and centered at the y-position of the source. In contrast, the centerline position of the plume in the z-direction changes drastically downwind from the source and
stabilises at approximately 300 m height at 1500 m from the source. As the plume gets mixed through the boundary layer, the variability in the plume centerline positions drops. Thus, while in the y direction the plume keeps growing throughout the domain, the growth stops in the z-direction once the plume reaches the top of the boundary layer (Fig. 9 c,d). The skewness of the plume positions (Fig. 9 e) shows that the plume is oscillating around its mean value in the y direction. In the vertical direction, however, the instantaneous plumes are more likely to have their centerline position below the mean plume centerline
in the first 1500 m from the source ($\overline{S_{za}} > 0$).





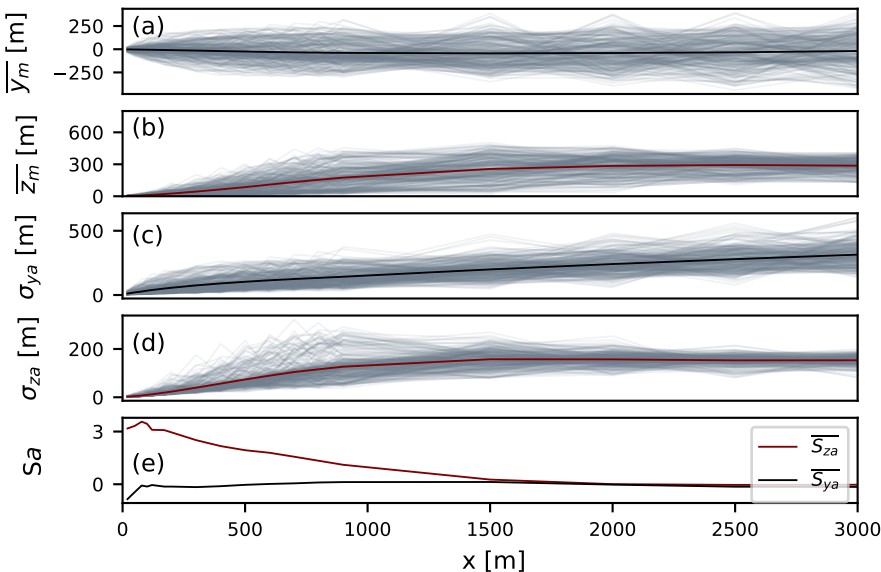

**Figure 9.** The first three statistical moments of the simulated plume in the absolute coordinate system. The first four panels show: position of the center of mass in y and z direction and the plume width in the y and z directions. All values are shown as function of downwind distance $x$. Grey lines denote instantaneous plumes and mean values are shown as solid black and red lines in y and z directions, respectively. The bottom panel shows the skewness of the mean centerline plume position as a function of distance from the source.

To inspect the skewness of the plume more closely, Fig. 10 shows probability density functions (pdfs) of center of mass positions in the $y$ and $z$ directions around their mean values at various downwind distances. As already noted in discussing Fig. 9, in the $y$-direction the plume positions show a Gaussian distribution on all distances from the source. Note that the spread of the centerline positions grows with distance from the source, indicating that the plume gets moved further away from the

mean centerline position with bigger and bigger eddies. This Gaussian distribution of the plume centerline position was also found in previous studies. For instance, Gailis et al. (2007) assumed a Gaussian distribution of the plume centerline position for their fluctuating plume model, which they experimentally confirmed in a water channel experiment. In contrast, close to the source, the centerline position distribution in the $z$-direction is positively skewed. A Gaussian distribution is attained further downwind. This result differs somewhat from the previous studies (Gailis et al., 2007; Marro et al., 2015), in which a lognormal

and reflected Gaussian distribution were assumed for modelling the plume vertical mean position. We show here that only close to the source, the lognormal distribution provides a good description of the centerline positions. Further downwind, where the plume gets better mixed in the convective boundary layer, the centerline position starts oscillating around its mean position.





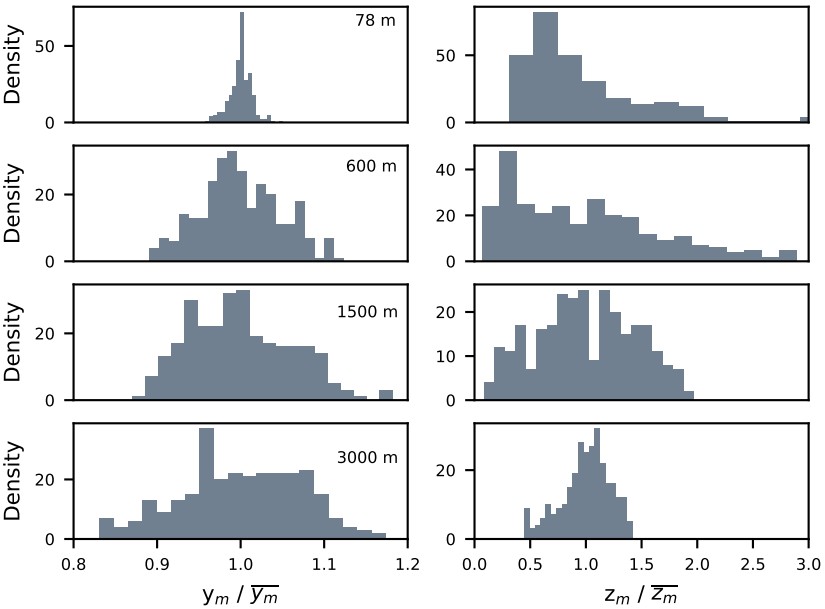

**Figure 10.** Probability density functions of the instantaneous plume position scaled with the mean centerline position at various downwind distances from the source.

### 4.2.2 Relative dispersion

There are two processes that affect plume growth: meandering motions (discussed in the previous section) and relative disper-
sion due to mixing by small eddies. Understanding these two processes, and quantifying where a certain process is dominant, can aid the development of measurement strategies. For example, at downwind distances where relative dispersion dominates, the instantaneous plumes remain close to the mean position, and the chance of measuring the plume close to its centerline increases.

Firstly, we focus on the relative dispersion. Relative dispersion is defined as dispersion of the plume around its centerline
due the eddies of comparable size or smaller than the plume. We present second and third order statistics of relative plume dispersion in Figure 11 (left row). Close to the source, the contribution of relative dispersion to the total plume growth is still small, which is especially visible in the $y$- direction (Fig 11 a). As the plume moves away from the source, it grows in size. This enables bigger and bigger eddies to be involved in the mixing of the plume. For this reason, the size of the plume due to relative dispersion is growing downwind from the source at a constant rate. Similar behaviour is seen for dispersion in the $z$-direction.
Close to the source, the contribution of relative dispersion is small and it grows further downwind. Unlike the $z$-direction, in which the plume growth is limited by the size of the BL, the growth of the plume in the $y$-direction is unrestricted by bound-aries. This implies that the plume growth will continue until the plume is so dispersed that it becomes indistinguishable from the background concentrations. The skewness of the relative plume dispersion in the $z$-direction shows a positive value close





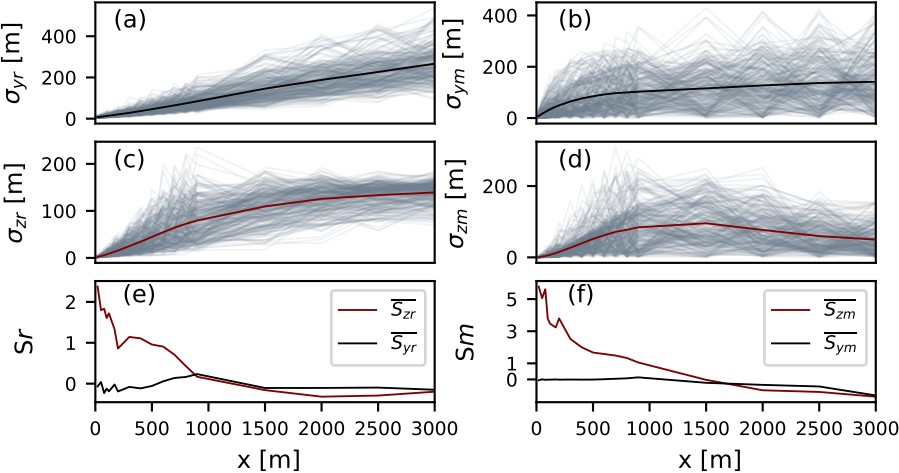

**Figure 11.** The second and third statistical moments of the simulated plume for the relative (left row) components centered around the mean plume position and for the meandering (right row) components. The first two panels in a row show the plume width in the y and z directions. All values are shown as functions of downwind distance. In grey are shown values of instantaneous plumes while their mean values are shown as solid black and red lines in y and z directions respectively. The last panel shows skewness of the mean relative plume position as a function of distance from the source.

to the source, which can be attributed to plume reflection at the surface. Since the small turbulent motions that are responsible
for relative dispersion are random and Gaussian in nature, the asymmetry has to originate from the fact that the plume is close
to the surface (Dosio & de Arellano, 2006). In the $y$-direction, the plume shows virtually no skewness.

### 4.2.3 Meandering

Finally, we look at the plume dispersion due to the meandering of the plume, i.e. displacement of the plume caused by the
eddies that are larger than the plume itself. Figure 11 (right row) shows the second and third moment of the plume due to
the meandering motions. It can be observed that in the $y$-direction the plume shows large symmetric growth very close to
the source (Fig. 11 b). The magnitude of the meandering component in this region is much larger than the relative dispersion
(Fig. 11 a). However, while the relative contribution continues to grow further downwind, the growth due to meandering drops
significantly and becomes almost constant. This observation is in line with the theoretical analysis given in Csanady (1973)
where $y$-scaling was reported according to $\sigma_{ym} = \sigma_v t$ close to the source, and $\frac{d\sigma_{ym}}{dt} = 0$ far from the source. This was later
confirmed in the water tank experiment by Weil et al. (2002), and the LES study presented by Dosio & de Arellano (2006). Our
results agree well close to the source. However, in our simulation there are still eddies big enough to move the whole plume
even far downwind, since our value of $\sigma_{ym}$ does not become fully constant. In contrast, the contribution of meandering to the
total dispersion in $z$-direction tends to zero further downwind from the source. The size of the eddies that develop in vertical





direction is constrained by the depth of the boundary layer. For this reason, with only meandering, the plume attains a size

comparable to these eddies.

The distance at which the relative regime becomes predominant can be estimated from the relevant convective and advective timescales introduced in Section 4.2.1. When the two time-scales become equal, the plume has spent enough time in "flight" to be mixed with the largest eddies. Therefore, a length-scale, $L_{mix}$, can be derived that defines the downwind distance at which the plume starts to be mixed with eddies of all sizes and at which the relative dispersion becomes predominant

$$L_{mix} = \frac{\overline{u} h_{BL}}{w_*}. \tag{9}$$


In this study, this distance amounts to $L_{mix} \approx 1360$ m. Alternatively, this distance can be obtained from the meandering ratio $M \equiv \sigma_{im}/\sigma_{ir}$, i = y,z (Oskuie et al., 2015). When $M$ drops to values smaller than 1, the relative dispersion becomes the dominant process. This occurs at $x \approx 1320$ m downwind of the source, which is in good agreement with the estimated length scale. Note that this distance is specific for each case. It depends not only on the turbulence regime and the BL height, but also

on the release height.

### 4.3 Concentration statistics

Finally, we will present concentration statistic in the absolute and relative coordinate systems. Additionally, we will compare these statistics to parametrizations that are commonly used in fluctuating plume models (Gailis et al., 2007; Marro et al., 2015; Cassiani et al., 2020). These fluctuating plume models have been validated against dispersion studies in laboratory channel

flows, often by taking line transects through the plume (e.g. Nironi et al. (2015)). Here we aim to utilize the high spatial and temporal resolution of LES to estimate dispersion parameters, needed in these models. Figure 12 (first and second row) shows y-z transects through the time-averaged plume (average of 287 instantaneous plumes) in the absolute (left) and relative (right) coordinate systems. In the relative system, the instantaneous plumes were aligned with the center of mass of the mean plume ($\overline{y_m}$, $\overline{z_m}$). It can be seen that close to the source (top row) the two plumes are similar in shape since the maximum of

concentration and centerline position still coincide. Further downwind, there is a clear difference between the plumes since the plume entered the regime in which it was more frequently carried upwards by strong ejections (see e.g. Fig. 10). On the distances furthest downwind (Fig. 12, bottom row) the two plumes again attain similar shapes. Here, the relative dispersion is the dominant mechanism and the centerlines of the instantaneous plumes do not move far from its mean position by meandering motions. For the two distances closer to the source, the edges of the plumes show large variability, despite the large amount

of plume transects that was used in time-averaging. This is caused by the plume behaviour. Close to the source, the plume tends to stay close to the ground. The large spatial variability away from the ground is caused by occasional ejections by strong upwards motions. Further downwind, the plumes attain a more uniform shape, resembling a Gaussian distribution.

### 4.3.1 Parametrization of concentration fluctuations intensity

One of the commonly used parameters to model the concentration *pdf* is the concentration fluctuation intensity, defined as

$i_c = \frac{\sigma_c}{\overline{c}}$ (Gailis et al., 2007; Nironi et al., 2015; Cassiani et al., 2020), with $\overline{c}$ being the mean concentration (x of plumes





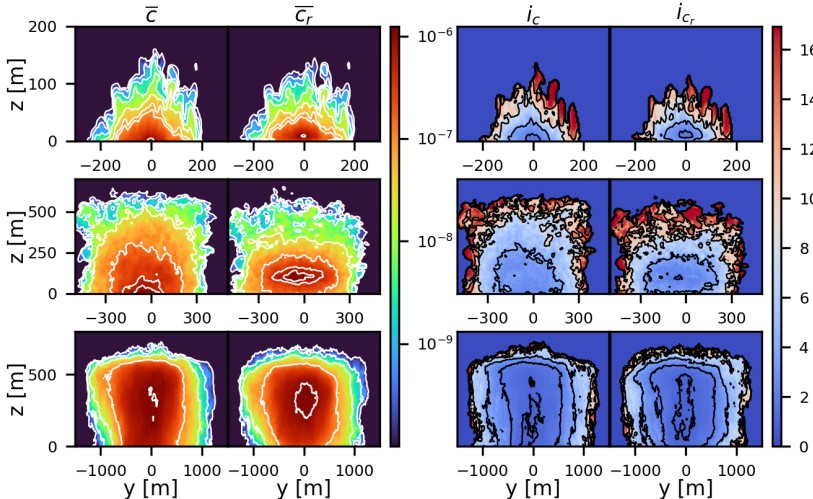

**Figure 12.** (First two rows) y-z transect of mean plume concentrations in absolute ($\overline{c}$) and relative ($\overline{c_r}$) coordinate system. (Last two rows) Concentration fluctuation intensity in y-z plume in absolute ($i_c$) and relative ($_{cr}$) coordinate systems. Distances from the source are (top) 100 m, (middle) 600 m and (bottom) 3000 m.

averaged) and $\sigma_c$ its standard deviation. As pointed out in Marro et al. (2015), the spatial evolution of this non-dimensional parameter is often assumed to depend only on the $x$-coordinate. This can lead to significant discrepancies between modeled concentration fields and the measured ones. We use LES to demonstrate the complex spatial structure of variable $i_c$, both in absolute and relative coordinates, over a plume crosswind transect (Fig. 12, third and fourth rows). Close to the plume

centerline $i_c$ has minimum value, but it increases noticeably towards the plume edge. This is most clearly visible close to the source. In the far-field, these differences are less pronounced, which is a consequence of plume being better mixed, which decreases the intermittent behavior.

Furthermore, from a measurement point of view, knowledge of the shape of $i_c$ can help in planning the measurement campaigns. High values of the $i_c$ imply that the probability of measuring the plume in that area are lower, longer measurement times are

required to achieve reliable plume statistics. Knowledge of the optimal downwind distance and height at which to measure can considerably improve the efficiency of the measurement process.

The measurements used in this study were taken as line transects at a single height (3 m). According to the results presented here, at 3 m height the plume was the least fluctuating very close to the source ($x \lesssim 300$ m) and in the far field ($x \gtrsim 1500$ m). We have previously shown that in the close field the pdf of vertical centerline position is positively skewed (Fig. 10 right row),

therefore there was higher likeliness of capturing the plume closer to the ground than at its centerline. Conversely, far from the source, the plume is oscillating around its centerline and there is the highest chance of measuring the plume. In the mid-field ($300 \gtrsim x \lesssim 1500$ m), the plume is highly oscillating at the ground and at the centerline position, but since it is still positively skewed, there is a higher chance of measuring the plume at the ground.





The complex 3D structure of $i_{cr}$ has been addressed in previous studies. Marro et al. (2015) expanded upon the definition of

$i_{cr}$ given in Gailis et al. (2007), where the relative concentration fluctuation has been expressed in terms of the mean relative concentration field. The model presented in Marro et al. (2015) is given as:

$$
\begin{aligned}
i_{cr}^2 = (1 + i_{cr0})^2 & \left\{ \exp\left[ -\frac{(y - y_m)^2}{2\sigma_{yr}^2} \right] \right\}^{-\zeta_y(x)} \\
& \times \left\{ \exp\left[ -\frac{(z - z_m)^2}{2\sigma_{zr}^2} \right] + \exp\left[ -\frac{(z + z_m)^2}{2\sigma_{zr}^2} \right] \right\}^{-\zeta_z(x)} \\
& \times \left\{ 1 + \exp\left[ -\frac{(2z_m)^2}{2\sigma_{zr}^2} \right] \right\}^{-\zeta_z(x)} - 1,
\end{aligned}
\tag{10}
$$

where $i_{cr0}$ is the value of relative concentration fluctuation at the plume centerline (Fig 13 a), $\zeta_y(x)$ and $\zeta_z(x)$ are the shape parameters introduced to account for anisotropy in the y and z directions. The variables that determine the crosswind shape of

$i_{cr}$, $y_m$, $z_m$, $\sigma_{yr}$ and $\sigma_{zr}$, need to be either determined from plume measurements, or parametrized using one of the models (e.g. Gailis et al. (2007), Marro et al. (2015)). Here they are calculated from the LES data as defined in section 3.3. The two $\zeta$ functions were assumed to be sigmoid, such that the modeled $i_{cr}$ has value $i_{cr0}$ close to the source and has self-similar profiles in the far field in both $y$ and $z$ directions. As previously mentioned, the LES data show the U-shaped profile in the far field, but also close to the source (not shown). This is likely caused by the fact that in the simulation the source is not introduced as

point source, as assumed in the plume model, but as a 2D Gaussian in the $x$ and $y$ directions with one standard deviation the size of one grid box ($\Delta x = \Delta y = \sigma_{source} = 5$ m). Therefore, 95 % of mass is being emitted from an area that has an horizontal transect of 20 m. The size of the smallest eddies that can develop in the simulation is $\approx 4\Delta x$. This means that very close to the source there is no internal mixing in the plume by the smaller eddies and all of the fluctuations are caused by entrainment of ambient air by eddies comparable in size to the plume.

We have adapted the definition of the shape functions $\zeta$ given in Marro et al. (2015) to account for the shape of the source in the near-field $y$ direction and kept the same behavior in the far-field. The far-field was defined as the distance at which the relative dispersion becomes dominant, therefore at the characteristic length scale $L \approx 1360$ m (section 4.2.3). The slope of the sigmoid function $\beta$ was determined using $L$. It was assumed that at distance $L$ from the source the value of $\zeta$ has $p$ % (here used $p = 70$ %) of the amplitude defined in Marro et al. (2015). $p$ was chosen as the ratio of relative to absolute dispersion for

the respective directions at the distance $L$. As a result, the functions take the shape:

$$
\zeta_y = \gamma + \frac{\alpha_y - \gamma}{1 + \exp\left[ -\beta_y(x - x_0) \right]}, \qquad \zeta_z = \frac{\alpha_z}{1 + \exp\left[ -\beta_z(x - x_0) \right]}.
\tag{11}
$$

Where $\gamma = 2 \times 10^{-3}$ is the correction for the shape of the source, $\alpha_y = 0.45$ and $\alpha_z = 0.9$ are the amplitudes taken from Marro et al. (2015), $x_0 = 0.5L$ is the location of the function's midpoint, and the slopes are calculated as:

$$
\beta_y = -\frac{2}{L} \ln\left( \frac{(1 - p)\alpha_y}{p\alpha_y - \gamma} \right), \qquad \beta_z = -\frac{2}{L} \ln(1 - p).
\tag{12}
$$

Figure 13 shows the comparison of $i_{cr}$ calculated from the LES data and $i_{cr}$ modeled with the two definitions of the $\zeta$ function. As previously mentioned, the definition of $\zeta$ found in the literature (Marro et al., 2015) agrees well with the LES-





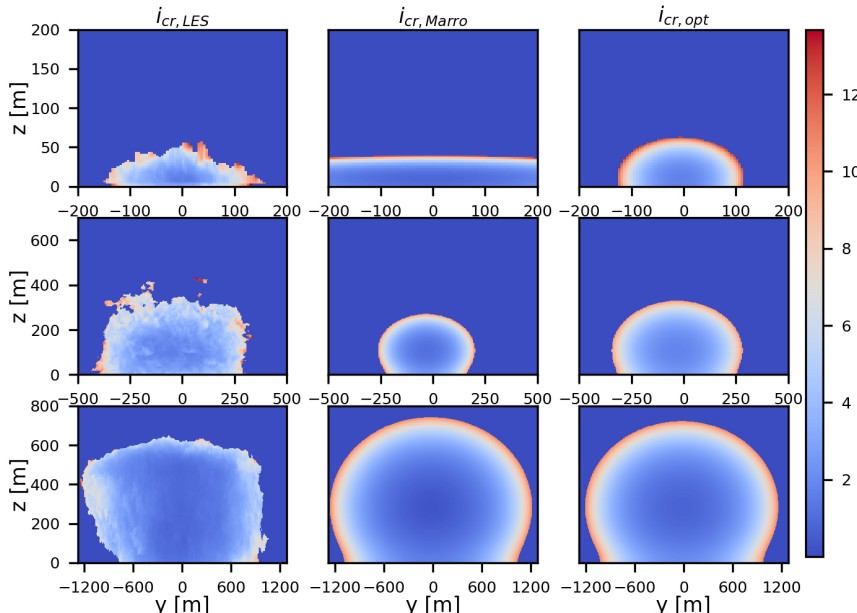

**Figure 13.** y-z transect of the concentration fluctuation intensity in the relative coordinate system $i_{cr}$ calculated from the LES data. The other two columns show $i_{cr}$ calculated with Eq. 10 as found in the literature (middle column) and optimized for this case (right-hand column).

calculated relative fluctuations in the far field. Very close to the source the, the LES plume has a similar structure as in the far field, which is not accounted for when the assumption of constant valued $i_{cr}$ is made. When the correction for the source shape is added (equation 11), the $i_{cr}$ model represents the plume behavior well, both in the far-field and close to the source. It should

be noted here that the plume behavior at distances from the source where meandering is important, is still misrepresented by the plume model.

One of the assumptions in the meandering plume model is that the relative dispersion and the fluctuations of the instantaneous center of mass are statistically independent processes. This assumption is violated when the size of the plume is comparable to the average size of eddies in the domain. In this case, the eddies that are capable of moving the center of the mass of the

instantaneous plume are still small enough to entrain ambient air deep into the plume making the separation of two processes complicated.

### 4.3.2 Concentration probability density function

Lastly, we look at the concentration $pdf$ at multiple in-plume locations. A large number of studies have found the Gamma distribution to be an appropriate description for the $pdf$ of relative concentrations in the far-field (e.g. Dosio & de Arellano

(2006), Nironi et al. (2015), Marro et al. (2015), Cassiani et al. (2020)). In the far field, relative dispersion becomes the main mechanism that drives the plume fluctuations. Therefore, the probability of the plume centerline position tends towards a Dirac





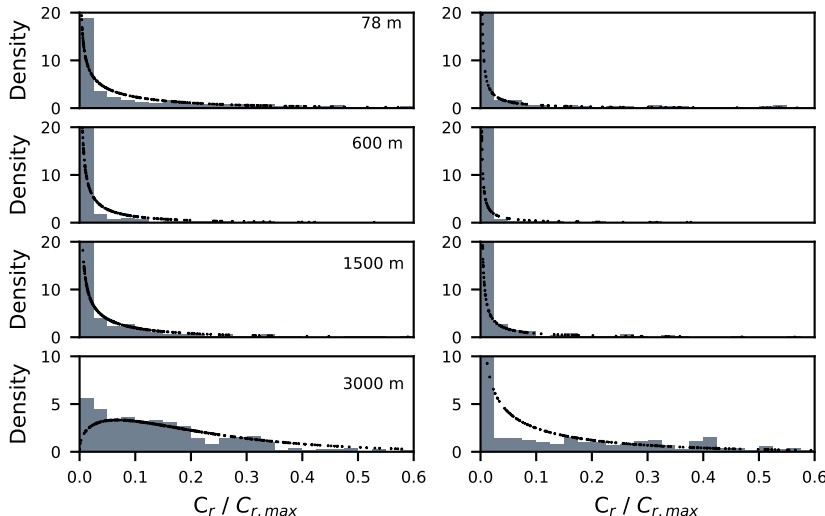

**Figure 14.** Probability density function of concentrations at the plume centerline (left) and at the measurement inlet height(right, see Section 2.2) at multiple downwind distances. Over-plotted are Gamma function fits.

delta function, and the plume spread due to meandering motions becomes negligible. The *pdf* can then be expressed as:

$$p = \frac{\lambda^\lambda}{\overline{c_r}\,\Gamma(\lambda)}\left(\frac{c_r}{\overline{c_r}}\right)^{\lambda-1}\exp\left(-\frac{\lambda c_r}{\overline{c_r}}\right), \tag{13}$$

where $\lambda = 1/i_{cr}^2$ and the subscript $r$ denotes the relative plume. Figure 14 shows *pdf*s of the relative concentration sampled at
the plume centerline on multiple downwind distances and, for comparison, at the inlet height at which data presented in section
2.2 were measured. The Gamma distribution is indeed a good fit for concentration *pdf* at the plume centerline on all downwind
distances. The Gamma functions also reasonably fit the concentrations away from the plume centerline (not shown). Very far
from the source (x = 3000 m), the gamma distribution poorly fit the relative concentrations at inlet height (z = 3 m). Note that
the $i_{cr}$ that was used here for the calculation of the *pdf* has been calculated from the LES data. We concluded earlier that the
$i_{cr}$ has a complex structure, which cannot be assumed constant in the y-z plane. When its value is known, either from data
or from an appropriate plume model, the *pdf* of concentration fluctuations can be modeled with a Gamma distribution for a
certain in-plume location.

## 5   Conclusions

Our study aimed to bring together field observations and high-resolution simulations. Large-eddy simulations (LES) have
been employed in dispersion studies for the past few decades, but most often simulating dispersion in somewhat idealized
settings. The models capable of performing LES are constantly being improved, with higher spatial resolution, and with new
parameterizations that include more processes that influence the plume dispersion. We demonstrated here the ability of LES
to reproduce plume dispersion in an actual field campaign. We took a step away from idealized channel flows, and used





available meteorological data to reproduce field conditions encountered during the campaign. Since field observations are sparse, LES can lead to improved understanding in plume behaviour, which can help with planning and optimizing future measurement strategies. The case we studied was a methane plume emitted from an oil well that was measured during one day of the Romanian methane emissions from gas and oil industry (ROMEO) campaign. The boundary conditions in the LES were derived from ERA5 data (Hersbach et al., 2020) to ensure correct meteorological conditions in the simulation. The plume in the simulation was released from the lower boundary and sampled in accordance with the field observations.

Firstly, the meteorological variables from the LES were compared with the available field data and the ERA5 profiles. The vertical profiles of specific humidity and temperature in ERA5 data showed little variability for the period in which the measurements were taken. The LES was able to reproduce these profiles correctly. There was very little large-scale advection present for the chosen day, which implies that the wind was driven by local temperature differences and orography that are not properly captured with the model resolution of ERA5. This resulted in discrepancies between the LES generated wind profiles and the measured wind. The issue was circumvented by applying a wind correction and performing a second simulation with this background wind. While the forcing of the boundary conditions with the ERA5 data gave good results, more detailed measurements of meteorological variables (e.g. vertical profiles of wind components, temperature, humidity etc.), together with plume measurements would help to better evaluate the simulations.

Secondly, the LES was compared against plume observations. A methane plume emitted from an oil well was sampled with an instrument mounted on a moving vehicle. A tracer gas plume emitted close to the oil well was measured simultaneously. The tracer gas plume was used in the estimation of the emission rate from the unknown source. Our aim in this study was to evaluate whether LES can be used as a proxy for the tracer gas. The estimate of the emission rate from the oil well using the tracer gas plume is $Q_{CH_4}$ = 1.23 g s$^{-1}$. Using LES, we found $Q_{CH_4,LES}$ = 1.11 g s$^{-1}$, i.e. a value 10 % lower. To further evaluate LES, we estimated the emission rate of the tracer gas ($Q_{N_2O}$ = 0.59 g s$^{-1}$) using the simulated plume, and found a value of $Q_{N2O,LES}$ = 0.53 g s$^{-1}$. Part of the differences in the estimated emission rates can be attributed to the different mean wind speeds in the simulation and in the measurements. Nevertheless, it was shown that, using a careful set-up of the simulations, LES can replace the co-emitted tracer gas, e.g. in cases of poor access to the source area.

LES provides concentration fields throughout the domain with great temporal and spatial detail. This allows for a more in-depth study of the behavior of the measured plume. The plume was studied by analyzing its absolute position, and by separating the processes driving its dispersion into meandering motions of the plume centerline and the relative dispersion around this centerline. A good agreement of the plume behavior was found with previous experimental and theoretical dispersion studies targeting channel flows. Furthermore, a plume mixing length-scale $L$ was derived from the boundary-layer height, the mean horizontal wind speed and convective velocity scale. This scale was demonstrated to coincide with the distance from the source at which the relative dispersion becomes the main mechanism of plume growth, and for this case study, $L$ is calculated to be 1360 m.

Finally, we used LES to examine parameterizations of concentration fluctuations in simple models: the fluctuating plume model. We did this by focusing on the concentration fluctuation intensity parameter, $i_c$, an often utilized parameter. LES can provide the detailed 2D fields of $i_c$, something that is difficult to obtain in laboratory experiments. We confirmed the



characteristic U-shape in a horizontal crosswind transect of concentration fluctuation intensity in a relative coordinate system
$i_{cr}$ (Gailis et al., 2007) not only in the far field, but also close to the source. We speculate that this is due to spatial extent of the
source in the simulation, imposed to avoid numerical instabilities. In this way the simulation differs from the field experiments,
where close to the source the plume is mixed by eddies ranging from the Kolmogorov scale to the size of the plume itself,
making the plume compact and very well mixed. We adapted the semi-empirical model for $i_{cr}$ from Marro et al. (2015) to
account for the source shape and this model showed good agreement with LES.

Furthermore, the knowledge of the shape of $i_c$ can help in planning future measurement campaigns as it is an indication of the
chance that the plume will be measured. For the campaign analysed here it seems that the plume was measured where there was
the highest chance of capturing it – close to the source and the ground. In general, far away from the source the plume is best
measured close to its mean centerline, which is likely lifted off the ground as the plume gets mixed throughout the boundary
layer. Close to the source, however, the plume is mostly below its centerline, so the chances for measuring it are higher closer to
the ground. Following the study of Dosio & de Arellano (2006) of dispersion form an elevated source in a convective boundary
layer, it seems that this is true for the lifted sources as well, close to the source most plumes first get transported to the ground
and then mixed through the BL with larger eddies.

Finally, previous studies found that the probability density function for concentrations in the relative plume can be described
by a Gamma distribution. This finding was also confirmed in this study. With the spatial variability of $i_{cr}$ is taken into account,
the Gamma distribution is a good fit for the concentration distribution on various downwind distances.

In conclusion, LES has shown to be an invaluable tool for studying plume dispersion. In this study LES has been pushed a step
further to bridge the gap between field experiments and simulations. LES can properly reproduce meteorological conditions,
but future campaign should provide more detailed measurements to further drive and evaluate the simulations. In the future,
more detailed LES models will become feasible due to more powerful computers. For this reason, high-resolution and realistic
atmospheric dispersion simulations will likely play an increasing role in tracer dispersion studies.

*Code availability.*    Simulations were performed using MicroHH model available at: https://microhh.github.io/

*Author contributions.*    AR, CvH and MK designed and performed large-eddy simulations. AH, IV and PvdB conducted the measurements
and provided the field data. AR wrote the manuscript in close collaboration with CvH, MK and AH. BvS designed the LS2D software that
was used to generate LES forcings from ERA5 data.

*Competing interests.*    There are no competing interests.





*Acknowledgements.* This project is part of the Methane goes Mobile - Measurements and Modelling (MEMO$^2$) project. This project has received funding from the European Union's Horizon 2020 research and innovation programme under the Marie Sklodowska-Curie grant agreement No 722479. Maarten Krol received funding from the European Research Council (ERC) under the European Union's H2020 research and innovation programme under grant agreement No 742798.





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
