# Peer review of "Technical note: Interpretation of field observations of point-source methane plume using observation-driven large-eddy simulations"

_Atmospheric Chemistry and Physics, 2021_

## Author Comment (AC1)

**Authors responses to review 1**

Firstly, we would like to thank the reviewer for their time and valuable suggestions for improving our paper. The review was very detailed and it is clear they spent a considerable amount of time to help us improve our text and we would like to thank them for that.

Secondly, we would like to stress to the editor that the design of the campaign was not done by the team that did this study, as the MEMO2 project has a separation between measurement and modelling activities. We agree with the reviewer that more detailed wind observations could have been really helpful, but unfortunately those were not collected. Nonetheless, measuring concentrations in the way that has been done in the field campaign is common practice in the estimation of sources, as often large areas need to be covered in the search for leaks. Hence, we still believe that our method is of great value for deeper interpretation and better source estimation.

In this document, we have repeated the reviewers comments in the *italics* and our responses are in the standard font.

*Personally, I would like to congratulate the authors of good work done with this paper. I think that a time spent for reading it is definitely not lost. However I'm surprised that this is submitted under the technical note format.*
We thank the reviewer for the kind words. The paper was indeed not written as a technical note. The re-classification of the paper into a technical note happened after the initial submission as a research paper after a discussion with the editor.

*The main topic of the article focus on the estimation of methane emission rate from OG installation in Romania during the campaign in year 2019. I hoped that application of complex LES model will be deeply interpreted for some good and bad examples of plume behaviour. Here, I mean the well known structure of the plume (I suppose that smoke would be much better medium to do the study than methane). The authors had another idea of the story, which is not bad but a bit hard to follow.*
The main idea behind this paper was to demonstrate the ability of LES to support the kind of mobile measurements described in the paper. Often times, due to road accessibility and the goal to cover as much ground as possible, these measurements have only a handful of plume transects at one downwind distance and perhaps wind data from a sonic. As the reviewer pointed out later in his text, such datasets are not very robust and the idea behind the paper was to fill in the missing information about the measured plume and also connect this real situation with semi-empirical models presented in other work, which was done later in the paper.

*So, at the beginning they present the description of measurement site and methodology, than touch a bit of LES set-up and some hints for further statistical analysis of gas plumes. All this., including introduction takes full 9 pages and not much details are presented anyhow.*
As previously said, the idea of the paper was to reproduce one day during the measurement campaign

with LES. We find it necessary to provide as detailed as possible description of the whole problem. From the meteorological situation, which ultimately influenced the measured plume, to the accurate measurement site description to the description of the actual measurement process and the instruments used. All of these, apart from the description of the instruments, are inputs for the LES simulation.

*The LES model (authors refers to the github repository of microHH) applicability should be tested on the known source during well designed experiment with the well measured state of boundary layer, including wind profiles and convection scales. This requires some extra instruments like doppler sodar, windcube wind profilers and perhaps a lidar.*

Agreed. To validate the LES for this specific case study more detailed measurements should have been taken during the campaign. However, this was the crux of one of the ideas behind the paper. Can LES reproduce the measured plume from insufficient data. The model itself has been validated independently from this study. The use of ERA data as a source of boundary conditions for LES has been utilized in Bosman et al. (2019, https://doi.org/10.1002/qj.3441) where they used ERA-Interim data to force boundary conditions in their study of cloud formation. MicroHH has also been tested for stable boundary layer conditions in the paper of van der Linden et al. (2019, https://doi.org/10.1007/s10546-019-00461-4). The ability of the LES to reproduce point source plume dispersion has been validated in Raznjevic et al.(https://doi.org/10.5194/amt-2022-25) against wind tunnel experiment of Nironi et al. (2015, https://doi.org/10.1007/s10546-015-0040-x) and has proven to reproduce the experiment very well.

*Authors were completely unprepared for the experiment and choose the one of the measurements done during the campaign instead of using the multitracer (instrument can measure almost all possible gases) release tests, which for sure they were able to organize.*

It is a bit unclear what the reviewer meant here. We indeed analysed the plume coming from the oil well with an unknown emission rate, but there was also a simultaneous release on $N_2O$ tracer which was measured and used here to validate the LES results. As we mentioned above, the LES study was organized after the campaign was finished. Therefore, the measurements could not be added. But the tracer was measured simultaneously with methane.

*If the finding of emission rate from a single gas well was a target of the article – it's not enough to give the impression of the method validation. In general it is hard to understand why do the authors use the oil well as the emission source. If the applicability of LES is a target – they should prove the success on many other cases/sources so the reader will get some statistical information. Finally, if the interpretation of LES is a target than lets run it in different scenarios of landscapes (flat vs steep, meadow vs forest etc.).*

We mentioned in our paper that it is written in the scope of the campaign targeted at oil and gas industry emissions in Romania. In these kind of campaigns often times measurements are scarce or incomplete (e.g. screenings with one transect per plume) since the goal is to cover as much ground as possible and measure a large amount of sources to get an estimate how much emissions there are in the whole measured region. In light of this, we chose one day of the measurement campaign in which the chosen plume was best measured and the tracer release was available. The goal was to show the ability of LES to reproduce the whole measurement day, from the meteorological conditions to the plume itself and the use of it to fill in the information about the plume that cannot be acquired from the measurements. We agree with the point the reviewer makes that the multiple sources should be included to show statistical robustness of LES. However, to achieve that a much larger dataset with information from multiple plumes in different location is required which is normally not available from this kind of campaigns. Furthermore, even if the dataset was available, conducting a series of LES experiments of that magnitude is computationally very expensive. Therefore, we refer to our comments from above where the validation of LES concerning the plume dispersion has been done on wind tunnel data. The final point the reviewer makes poses a very interesting research question that would bring many useful insights into the plume behavior in realistic settings. This, however, encounters the same problems as above as it requires carefully planned measurement campaign for validation and also considerable computing capacities.

*I will try to express my point in a detailed review, but before I do that I would like to underline that I spent some time in reading and perhaps didn't follow the all authors suggestions and I put much time to help authors make the next, better approach to final text:*
We thank the reviewer for their effort and time. We address each of the detailed comments separately and where appropriate we implement the changes in the text.

*Line 74 − 81 : nice summary of the paper but it was just described in lines 59 − 73.*
Yes, we give first the motivation for the paper and then shortly describe the structure of the paper for easier navigation through it.

*87 What is the particular name of plain and mountain ridge where the experiment took place, Carpathians are 1700 km long.*
Oil well 1474 in Darmanesti, in the region between the Transylvanian Alps and Bucharest. Added in text.

*87 is "actual site" a single oil well? If this would be the very well measured oil well I would feel that its OK but later authors declare that it was just multiple transects on one road in some distance from the well (repeated cross-sections in same place). In this case we really use the cannon to shoot the sparrow.*
Yes, it is the single oil well with an unknown emission rate. The plume was indeed measured by taking transects on a road downwind from the source. While the well itself was not measured, a tracer gas was released from right next to it and measured simultaneously. We hope this way there is less splatter from the sparrow.

*89 and 90 and Fig1 − If length of the road is 150m and the bisection of the segment is not passing the source, it means that the mean distance between source and receptor is NOT the distance to the middle of the road segment.*
Yes. This is correct. We corrected it in the text. We also exchanged the sketch on Fig. 1 for the Google Earth view of the site.

*Fig 1. What for the North direction is reported? Average wind direction would fit better. The wind direction during the measurement was 50 till 150 degrees (let's say on average 100), so the road was mostly not in the central plume. It would require wind direction of 120 till 200 degrees. Wasn't any better point for the presentation?*
The figure was presented in the North - East coordinate system for the convenience. The road on which the measurements were taken is the only access road that the car with the measurement device could be driven on. After additional consideration, we found the Google Earth image of the site helps better with the description than the sketch that was first used.

*Fig 2. A) why the pressure situation is presented for a whole Europe – its just enough to present the region with large distances between isobars (here geopotential is not necessary as we discuss the low winds only). B) measurements of heat fluxes require basic meteorological instruments, there is no need to use the model when real data are easy to access. The experiment was obviously not planned during the campaign.*

A) The pressure situation over Europe was presented in order to give an overview of the very stable conditions that were present in Southern and South-East Europe at the time this campaign took place. However, we recognize that the pressure situation over the whole Europe is unnecessary and the text has been re-written to focus only on the E and SE Europe and Fig. 2 a) has been changed. Our goal with this discussion was to give a synoptic background for a case when a full field campaign is not available. We use our simulation to to fill in possible missing information in conditions of sparse data, i.e. not a full campaign but some data to validate the model with.

B) The model data was used because there was no measurements available.

*95 – 98 the synoptic situation is not referring to the Fig2a. as there is no high pressure over the Baltic sea visible. All the story can be shorten to sentence in lines 98 – 100. Fig2a is a bit useless.*

It is not written in the text that there is a high pressure over the Baltic, rather that there are large gradients over that region. As mentioned above, Fig. 2(a) has been changed to focus only on the area of the campaign.

*104 – 105 The windcube information would be welcomed for such experiments so the wind vertical gradient can be verified experimentally. Again, no need for comparisons of models if only experiment is well planned.*

We agree that the vertical profiles from the wind cube would be preferable in this situation. Unfortunately, they were not available. However, due to the very stable synoptic situation shown on Fig.2 we assumed the ERA5 data with its relatively coarse resolution still represented the real conditions quite well.

*111 "...LARGE scale advection was SMALL..."*

This is an unfortunate style choice which was changed in the text.

*120 Concentration is not referring to ppb value but any unit per unit of volume. Use the molar fraction instead.*

Yes. We changed the text.

*121 What does B20 means regarding the flask type?*

B20 flasks are 20L cylinders; volume at 100bar = 2000L gas. Added to text.

*122 NOAA (WMO) scale doesn't cover 5000ppb, ICOS scale as well. Please describe how does the linking procedure looks like or explain why it is not important.*

The concentrations of the working standards used for the calibrations were measured at the Cabauw Tall tower in the Netherlands, which is calibrated with the Integrated Carbon Observation System (ICOS) & the National Oceanic and Atmospheric Administration (NOAA) standards. Besides that, during the Romania Campaign, calibrations cylinders from Utrecht University were used, with concentrations of 6.3, 27 and 130 ppm for the higher concentration measurements.

This description was added in the text.

*122 The accuracy of measurement is the important parameter, please provide, including all internal uncertainties (calibration, averaging, etc).*
The instrument measured $CH_4$, $C_2H_6$, $N_2O$, $CO_2$ and CO simultaneously at 1 Hz with a precision of 2.4, 0.1, 386.3 and 2.5 ppb, respectively. Here, precision is reported as three times the standard deviation of six minutes' constant concentration reading.
Added into the paper.

*158 Why domain was chosen so large(almost 5km) and horizontal resolution was chosen to be so low (5m), Can MicroHH model be run on resolution of single meters?*
The plume dispersion is driven by the processes on a whole range of scales, from Kolmogorov to the very large eddies that drive the meandering part of dispersion. In order to have these large eddies develop, and which are created on scales of $\approx$ km, the domain of the simulation has to be large enough. Therefore, in order to capture most of the relevant length-scales that affect the plume dispersion, not only close to the source where the measurements were taken, but in the far-field as well, we have opted for the larger sized domain. This of course, comes at a price for resolution where we have chosen somewhat coarser grid to keep the already considerable computational cost from getting too large.

*162 The oil well is usually less than 1m wide (area max =3m2)and in LES model it is the area of 10x10m = 100m2? If yes, than it is far from realistic situation. Maybe other oil or rather gas facilities would be a better choice.*
This, we are aware of and is unfortunately a consequence of the choices we made for the domain size and the spatial resolution. For this study we had to choose from a set of measurements of a whole array of different sources measured during the ROMEO campaign. We chose the one that was best measured. This means plenty transects through the plume were taken, the wind was measured and there was a tracer released. Unfortunately for us, that source was a very small oil-well.
Small correction to this comment is that the source was implemented as a 2D Gaussian, so most of the mass was emitted from the surface area of 25 $m^2$, which is still much larger than 3 $m^2$.

*176 initiated FOR -¿ initiated AT*
Fixed.

*175 and 177 Simulation was run for 7h or 7.5h? Nudging was started 0.5h after the model started?*
Yes, this is fixed. The simulation was indeed run for 7.5 hrs and the nudging to the ERA5 profiles started at the begining of the simulation.

*183 what does "local" refers to? What are the explicit "local influences" and how they were represented in LES?*
The meaning of the sentence in the paper was that the first simulation resulted in the wind that showed very high fluctuating behavior when compared to the observations. Considering the synoptic situation discussed heavily in the paper, but also in this document, we concluded that the mean wind direction that was present during the measurements was a consequence of some local influences that are not captured in the ERA5. Since the domain in the simulation corresponds to the area around the oil well, which was a flat grassland, we centered our mean wind direction and repeated the simulation. To be able to capture any of the possible local influences, i.e. a slope flow, a much larger domain would be needed with the corresponding orography inside. To capture the plume, the resolution would have to stay the

same, which as is is on the coarser side as the reviewer pointed out previously, and at this point such a simulation is unfeasible.

*186 "all other specifics. . . .identical" – Table 1 points differences in columns 4,5,6*
Yes, the tendencies of the $u$ and $v$ wind components in the new simulation were turned off, as was the geostrophic wind and the location of the source was changed in order to capture the largest extent of the plume possible. In all other aspects the simulations are the same.

*192 "realistically" – what is this term referred to? Please explain why wind speed at 2m force the whole BL to be realistically simulated. Especially in case of "local influences" like the slope flow.*
Realistically in the sense that we aimed to match 2 m wind speed in the simulation to the measured one. The rest of the vertical velocity profile unfortunately we cannot verify against any measurements. The vertical profile shows a well developed mixed layer above the surface layer and that is expected for the general conditions like there. The local influences, again, cannot be verified.

*Fig3. What is the message from this profiles, especially for wins speed measured at 2m above ground and receptor located aprox. 100m from the source?*
Similarly as with the discussion for the synoptic situation given above, we wanted to show the full vertical profiles as they are what ultimately drives the wind at 2 m height and at the receptor 100 m from the source.

*199 is uref=umeas? Why is it differently subscripted? Why authors didn't use the simple Gaussian model for a comparison as it is frequently used tool in such situations? Why authors didn't refer to*
No, $u_{ref}$ is not the same as $u_{meas}$. As it is written in the paper, *ref* refers to measurements of the reference plume, either be it a simultaneous tracer released next to the unknown source, or a modeled plume. In our case that was a plume in LES, but it could be a Gaussian plume. We did not use the Gaussian plume because we wanted to offer an alternative to it since the Gaussian plume requires many simplifying assumptions on the plume and the flow field that add to estimation errors. It has been used in studies similar to this one see Caulton et al. (2018, https://doi.org/10.5194/acp-18-15145-2018), Raznjevic et al. (https://doi.org/10.5194/amt-2022-25). Also, for our study a tracer release from experiment was available as a validation for the method. There was no need to use the Gaussian plume.

*Fig 5. It is not explained in the text nor in the title of this figure why and how was the cross-section of plume chosen. Why at 3m? Receptor was at this height, right? So why wind measurements were done on 2m height? The source is located at 3600,3600, why not to show larger part of the plume and if it is settled up for the measurement scale – why not to show the transects (i.e. road segment). Why time between snapshots are not equal (nether the less it is not important).*
Yes, the receptor was at 3 m height and that is why the plumes shown on Fig 5. are at that height. This has been added to the text. As was described in section 2.2 the gas receptor was placed in the car that was driving on the road. The wind measurements, however, were taken with Gill R3 sonic anemometer which was placed close to the source. The anemometer was placed on a pole of 1.8 m, which is part of the standard equipment.
On Fig. 5 we show the plume up to 400 m downwind from the source. As this figure was intended to

illustrate the fluctuating behaviour of the plume that prompted us to set-up an additional simulation. We found it superfluous to extend the figure far beyond the extent on which the measurements were taken. Especially since on such a figure the details around the source would be lost.

Figure 7 shows transects on the road segment, albeit time averaged. We found no additional information from showing the instantaneous transects would be gained.

The time between the snapshots is now equal.

*233 Authors claim that "local effects..." not mentioned what exactly they introduced to keep the local condition "realistic" beside the (somehow) averaged wind speed.*

textitRealistic refers here to the simulation with the nudging according to the ERA5 profiles exactly, that means the wind too. Fig. 5 is there to illustrate that the wind direction is highly variable in this simulation. This is not in agreement with the wind measurements from the campaign. From this we concluded that the wind during the measurements was driven by some "local effect" we had no information on since ERA5 is too coarse and there were no measurements available. For this reason we matched the wind in the *idealised* simulation to the measurements and conducted the second run.

*248 How exactly release of N2O is measured? Give the uncertainties or accuracy estimation.*

N2O was released using a critical orifice of 0.65 mm2 at 5 bar. Before and after the release, the mass of the cylinder was determined. Release was 0.59 +/- 0.02 g/s. This is added to the section that describes the instruments.

*251 How one can tell the periodic behaviour of period 55min from 90min time scale figure?*

The wind data from the campaign was recorded for the whole duration of the measurements. That means from 11.30 UTC until 14.30 UTC. The periodicity was estimated by doing a Fourier transform on that data. But as was mentioned in the text, that was only speculation on our part and to actually confirm this claim a much longer timeseries is needed.

*Fig 6. A) what does 17 means on the time scale description. Why A) is in different timescale than C)? is B) plot on other timescale? If not you can bond the plots and use only one scale. Is legend at C) referring to B)?*

17 is the date, 17 of October. It is removed now from plots.

A) is in different timescale than B) and C) because we wanted to show how the wind in the LES compares with the measured wind. We indicated in the text that around 14.00 UTC the surface fluxes become negative and that is when the simulation stops. Therefore we are showing the wind on the timescale that there is data for the both simulation and the measurements. We show all of the plumes measured during that day, i.e. from 11.30 to 14.30 UTC, on A).

The x-axis on the plot is now shared and the legend in C) also refers to the B) plot.

*252 It is a bit of speculation with no reference, some link to the observations of orography influence on period of wind speed should be added.*

Yes, as we mentioned above that it was a bit of speculation on our part since we need a longer dataset to make any definite claims. However, we refer to the paper of Nastrom et al. (1987, https://doi.org/10.1175/1520-0469(1987)044<3087:AIOTEO>2.0.CO;2) where they have shown that the mountains can have influence on atmospheric variability extending 4 to 80 km. The distance to the closest mountain at the location site is well within that range. We added the reference to the paper.

*258 At what frequency wind was rotated, to average values or each 1min average of measured wind dir, if other please specify.*

To the average value. Added to the text.

*259 What measure does the "comparability" has? Be precise.*

The standard deviation of the wind directions on Fig. 6 c) is for the measured wind $\sigma_{WD,meas}$ = 16.9° and for the LES $\sigma_{WD,LES}$ = 18.6°. The two had very similar wind direction variability. This has been added to the paper.

*261 What was the time averaging period for mole fraction measurements, was it 20Hz or 1Hz or 1min as well? Not specified in "measurements" section.*

1 Hz. Added to the paper in the measurements section.

*265 There is no information during how many transect no plume should be observed according to the wind direction. Was this percentage in agreement to LES? Assuming 1 transect every min – we have 10% success for first 30 min and a bit more, up to 30% for the next periods. Is this poor recovery in agreement with LES?*

. With the way we have set-up our LES, there is always plume observed at the transect that corresponds to the road on which the measurements were taken. As we mentioned above, the mean wind direction in LES was directed perpendicular to the road. The only way a plume would not be measured there in LES would be if a wind sweep would lift the whole plume off the ground and the plume would be above the 3 m height.

*269 Fig2 shows the heat flux, not the surface flux of potential temperature – it is the same but please harmonise the variables*

Harmonized.

*271 "…looks smoother…" is not technical nor quantitative description*

The average skewness of the measured plumes are $S_{N_2O}$ = 0.42 and $S_{CH_4}$ = 0.44 for the $N_2O$ and $CH_4$ plumes respectively. However, the LES plumes have the average skewness of $S_{LES}$ = 0.32. Skewness here is defined as $S = \frac{1}{\overline{x_{max}}N}(\sum_{i=1}^{N}(x_i - \overline{x})^3)^{\frac{1}{3}}$, where $x$ is the plume transect from either measurement or simulation, $\overline{x}$ is the mean value of a transect and $\overline{x_{max}}$ is a maximum value of the respective mean plume (maximum of red line on Fig. 7). From this it is visible that there are less extreme values in the mean LES plume than in the mean measured plumes and this is a consequence of averaging as the set of LES plumes is larger than the measured sets. This has been added to the text.

*272 – 273 Why not to take only this simulation which potentially is able to be recorded by the receptor*

We believe some misunderstanding happened here. Fig. 7(c) shows results from one and the same simulation. So the half hour averages there are from the same simulation. Due to the differences in the wind direction, we presented the results as half-hour averaged plumes with the mean wind direction in that half-hour in the inset on the figure. The plume in the simulation is always going to be visible in the results as we have taken the transect (at the height of the receptor and at the corresponding downwind distance) over the whole width of the domain. Therefore, unless a wind sweep lifts the whole plume above the chosen transect height, there is always going to be plume measured.

*275 "...by 50" is not in agreement with line 264 where we find 40*

The wind during the measurements deviated by 40° from the mean during the first half hour of the measurements, so 11:30 UTC to 12 UTC. The wind in the simulation deviated by 50° from its mean in the last half hour of the simulation, so from 13:30 UTC to 14 UTC.

*276 – 284 – The averages of measured and simulated methane source efficiency don't contain uncertainties – whole comparison is qualitative – what is not welcome in technical notes. One can't compare the values.*

Uncertainties, estimated as a standard deviation of estimations from the half hour profiles, have been added to the text.

*284, 285 "...contribute to the error", "...estimation error might..." – do authors refer to uncertainty, difference between the results or real computational error? In all cases it should not be left unspecified. Also quantitatively!*

The errors we are discussing in that part of the paper are referring to the difference between the emission rates obtained from the simulations and measurements and the real emission rate. The lines have been re-written so that this is clearer. They have been expressed quantitatively now.

*286 It is not clear what "...not Gaussian shape..." is referring to and how authors made this statement – none of the curve on all 3 parts of Fig 7 looks Gaussian but there are some statistical test to make such statement, even if human brain is a powerful tool for curve recognition.*

We agree that none of the curves on Fig.7 are Gaussian. We have now added results from the Shapiro-Wilk test to show that. For the three curves that represent the mean plume over the whole experiment shown in Fig. 7, the p-values are $p = [9 \cdot 10^{-6}, 6 \cdot 10^{-16}, 1 \cdot 10^{-16}]$ for the LES, $CH_4$ and $N_2O$ plumes respectively. Therefore, the measured plumes are deviating from the expected Gaussian profile and are not as smooth (in reference to the comment above) as the LES one due to the much smaller set of plumes being averaged.

*289 Is it possible that from this section on all material is not referring to the earlier conditions?*

No. From the section 4.2.1 on we give deeper look into the LES plume and move away from the measurements. The idea is that the LES plume is a good proxy for the measured $CH_4$ plume and from the LES the additional information, unavailable from the measurements, can be inferred. So the same simulation as before.

*297 How does the averaging works for the plume this time? Is it 30 min average?*

The plumes have been averaged over 105 2D transects (x-y or x-z) taken over the whole duration of the experiment. This has been added to the paper.

*How do the integration works – is it same as averaging or rather adding up the plumes? Width and depth of plume are the dimensions not a cross-sections of plume.*

The integrals refer to summation over a given axis for each timestep. For example, Fig. 8 (b) shows the x-y cross-section of the time averaged and integrated plume where the integration has been done over the z axis. Width and depth are referring to the direction of integration, so for x-y transect the plume has been integrated over its depth, i.e. z direction, similar for the plume x-z transect there the integration has been done over the y direction.

*Fig 7. The N2O and CH4 averaged plumes are very similar however there is no quantitative measure to c confirm that from the figure. If the mass fraction (why mass fraction is used instead of molar fraction?) would be scaled or normalized to emission rate than figure would be more informative.*

We have scaled the plumes in Fig 7. with their respective emission rates. For the scaling of the methane plume, which is the only one with an unknown emission rate, the emission rate calculated using the tracer is used.

*315 Again authors do not present the measure of dispersion or uncertainty of values. How hBL is calculated so precisely (congratulations!), it refers to particular moment or whole period (11:30 – 14:30). Usually in this time it is rising slightly due to entrainment and heat fulxes, maybe not much but definitely larger than 1cm.*

The given values have been taken from the simulation. The boundary layer height has been calculated as a the maximum of domain- and time- averaged vertical profile of potential temperature gradient. This quantity is easily obtained from the MicroHH output. It is also easily visible in vertical profiles of potential temperature on Fig. 3.

In the paper we give the time in UTC, however Romania is 3 hrs ahead of UTC. Therefore, 11:30 – 14:30 UTC on October 17th is 14:30 - 17:30 local time. In that period of the afternoon the BL height is usually either stagnating or lowering due to the diminishing of heat fluxes.

*Fig 8. The scales on the plot affect the plume shape very much. Here X and Y scale are different while X and Z are the close. It gives very wrong impression that plumes are narrow. Also the integration of the plumes gives the wrong effect of plume density (what is especially important for instantaneous plumes). Taking into account the aim of the paper – cross-section would be more appropriate.*

The integration of the plume might give the wrong impression of the plume density. However, we find it is a very nice tool to show the 3D structure of the plume which is an advantage from simulations like these and is also important for the rest of the paper. The instantaneous non-integrated plumes can be seen on Fig. 5. Despite the simulation shown on Fig. 5 not been used due to the highly fluctuating wind direction, the structure of the instantaneous plumes differs little in the two simulations.

The height ratios between subplots on Fig. 8 have been changed. The width of the plume is shown more intuitively now.

*321 What "…starts" mean, how it was estimated and how it refers to value 1.9 (what are uncertainties?).*

By "starts" we meant that the position of maximum concentration is converging to the mean plume centerline. By converging we mean that its first derivation is monotonously approaching zero. This is mainly an observational result as the position of maximum concentration is quite noisy so it is difficult to discern any distinct limit at which it changes behaviour. Even with different smoothing techniques applied. This is the reason the approximation is used instead of an equality sign.

*324 Is top of BL 500m this time?*

The top of BL, $h_{BL}$ = 564 m is estimated from the domain- and time- averaged vertical profile of potential temperature gradient, as mentioned above. Assuming the comment here refers to the Fig. 8 (c) which is a snapshot of y-integrated instantaneous plume. Therefore some difference between the $h_{BL}$ is one time instant and $h_{BL}$ averaged over the whole simulation are to be expected.

*326 It is hard to judge but on Fig 9 there is much more plumes than 30*

It is not clear to which part of the text this comment is directed at. The gray lines on Fig. 9 represent one of the first three statistical moments (depending on the panel) of instantaneous plumes from the whole simulation. 287 plumes have been used to generate the statistics shown on Fig.9.

*335 From fig 8d the reader gets impression that the maximum of methane mole fraction stays deep below the mean of the plume centrelines till 2000m – here the skewness suggest 1500m, could authors give the deeper explanation of the difference. It is very doubtful however that someone would measure the oil well methane plume 1.5km from the well.*

The skewness is defined in terms of the difference between the instantaneous plumes center of mass and the center of mass of the mean plume (section 3.3). However, distribution of concentrations in an instantaneous plume can also be skewed from its center of mass. This can be seen from the time averaged plumes on Fig. 7 which are not Gaussian. Once the plume gets well mixed and the position of center of mass coincides with the position of the center of mass can the two figures (Fig. 8(d) and Fig. 9(e)) be compared.

*Fig 9. Scales on the figure are also different for each of the space dimension. It makes the plots equally wide but one can scarify it for sake of reality. Also Y distribution and horizontal shape of plume would be more visible. The first 50 - 100 m there is a negative skewness of y distribution, it is important from the point of view of measurement done in this region. Authors should comment on it. Why fig 8 has X distance up to 4320m and this one only 3000?*

It is unclear what the first part of this comment refers to. Apart form the skewness and the $\sigma_{za}$ plots, all the plots have y dimension in the order of $\approx 500$ m and equal x dimension.

It is true that very close to the source the position of the plume centerline is slightly negatively skewed. However, this skewness is not large (-0.8 closest to the source at 78 m) which is also visible on histograms shown on Fig. 10. We suspect this skewness is originating from the fact that longer averaging times are needed to smooth out the influence of the outliers (caused by possible turbulent structures with longer lifetimes that may have developed in this flow) in higher order statistics. We do not expect this skewness to be seen in similar experiments.

The distance on Fig. 9 goes to x = 3000 m because all the statistics have been done on y-z transects through the plume which have been taken on 18 downwind distances from the source. The transects have been recorded for the full duration of the simulation, so in order not to make the model output too large the number of transects had to be limited. We chose to have more densely distributed transects closer to the source where we expected for the plume to undergo larger changes. Further away we assumed the statistic of the plume would not change much, therefore there was no added value in taking transects all the way to the end of the domain.

*338 First approx.70 m has a skewness (negative), so not all distances are Gaussian shape. From the data presented in the paper it is not clear than all transects of the plume are Gaussian shape – it should be tested and results presented on the plot.*

As discussed in the previous comment, we agree that close to the source there is slight negative skewness in the distribution of the center of mass positions (Fig. 10 (a)), but we believe this is a consequence of insufficient averaging that would smooth out high values close to the source and that is not something that would occur in different situations. However, to be fully confident in this claim we would have to

perform another simulation for a different day with similar conditions which is computationally very costly.

*340 Even small eddies will make the plume wider with distance. What is the background of sentence "....with bigger and bigger eddies."*

The meaning of the sentence is that very close to the source the plume is very compact and it has the shape very similar to the shape of the source. There the plume is being moved by eddies larger than itself, or meandering, and those eddies do not change the shape of the instantaneous plume. Eddies of sizes smaller and comparable to the size of the plume can entrain clean air into the plume and make the instantaneous plume expand around its center of mass. As the plume moves away from the source, and grows, the range of eddies that cause the meandering reduces and the range of eddies that stretch and grow the plume around its center of mass grows. This is nicely summarized in the paper of Cassiani et al. (2020, https://doi.org/10.1007/s10546-020-00547-4).

*335 − 345 Quantitative description should replace the qualitative one. The oscillation of single centreline is induced by the eddies but its not obvious that it will result with oscillation of sigma y as well. Shouldn't the turn of the plume make it narrower not wider?*

In the indicated paragraph we do not claim that the oscillations of the plumes center of mass will instantly mean oscillation of $\sigma_y$ as well. That paragraph discusses the center of mass of the instantaneous plume only. However, from the definition of the plume spread $\sigma_y^2 = \sigma_{ym}^2 + \sigma_{yr}^2$ Cassiani et al. (2020, https://doi.org/10.1007/s10546-020-00547-4) it is visible that the oscillation of the plume centerline contributes to the total plume growth through $\sigma_{ym}$ as the total mean plume spread is a combination of contributions from the meandering and the relative dispersion ($\sigma_{yr}$).

*Fig10. The distribution of Z plume dimension 1500m from the source is not in accordance with fig 8 but is in accordance with fig 9 (are this figures from same simulation groups?).*

All of the figures in this paper are produced from the same simulation. Figure 9. (d) is showing the plume spread around its mean centerline (Fig. 9 (b)) which at the distance 1500 m is no longer close to the ground. If they are added together then the dimensions visible on Fig. 8 are obtained.

*355 It is not clear which dimension of plume authors mean (X,Y,Z)?*

y and z dimensions.

*358 Again the conclusion of "bigger and bigger" is not confirmed*

The "bigger and bigger eddies" refers to previous research. We added a reference.

*Fig 11. "...(left row).." refers to left column and next to the right comlumn*

Indeed. It is fixed now.

*366 What means "virtually"?*

Barely any.

*369 The dimension of plum might be also in X dimension. Replace "row" with column.*

It can, but in our paper we focused only on the plume growth perpendicular to the mean wind.

Replaced.

*371 "...is much larger..." can it be expressed quantitatively?*
Its 2.5 times larger at the highest difference. Added to the text.

*374 – please explain the variables , sigma v and t was not introduced earlier*
$\sigma_v$ is the variance of the $v$ component of velocity and $t$ is the time since the plume left its source. It is in the text now.

*379 – 380 Can it be confirmed? The size of distribution and size of eddies not necessarily might be corelated. Smaller eddies may make the overall plume spread by bigger distance with time (Csanady equation) its not obvious that vertical scale boundary induces other eddy-plume relation.*
We are not sure we understand the comment fully. The the reviewer is referencing is discussing the fact that the largest eddies that can move the plume, in vertical direction, are constrained to the depth of the boundary layer. Therefore, the mean plume growth from meandering is also constrained to the size of those eddies. As the plume moves away from the source, it slowly grows and the range of eddies that contribute to the spread of the plume around its center of mass becomes larger. So we are not contradicting Csanday here.

*387 – Y variable is not important as we are describing the vertical distribution only (equation 9 is only referring to Z)*
It is not indeed. Nevertheless, it can be defined for y as well.

*388- 390 Usually sources of methane related to OG are not higher than few meters. It will not change the picture. Authors do not present any simulation for higher emitters nor the thermally elevated plumes.*
We agree with this comment, we do not present here analysis for higher emitters or for thermally elevated plumes. However, the analysis we present in this paper can be applied to any point source plume released from the ground in similar atmospheric conditions. It does not have to be only methane plumes from O&G. That is why we point out that in the length-scale parameter $L_{mix}$ discussion that it is not applicable for different flow types and different emission heights.

*Paragraph 4.3 has no quantitative results from modelling. I would expect it from technical note.*
Paragraph 4.3. motivates the last part of our paper through short reference to previous work and through Fig. 12.

*Fig 12. In title please exchange "rows" with columns. Dimensions (Y and Z) of the plumes are not consistent (even not consistent ratio) – as the shape, moments and distribution of methane the topic of the article it makes completely illusive picture for a reader.*
The reviewer is right, the figure is not consistent in dimensions and ratios between the rows of subplots. This is because the figure shows the plume at different stages of its growth, were all of them plotted on the same scale details in the plume closest to the source would not be visible and we find that defeats the purpose of the figure. We traded off the realism for details. The axis ratios can be adjusted but we are also constrained by the requirements on the figures by the journal itself, so we cannot increase the size of the last row in the figure and if we shrink the z axis to scale the figure will become unreadable.

Therefore for the sake of readability and comparability, we decided to have the figure in this form. The caption is fixed.

*415 the definition of ic implies the shape of it. Authors don't propose any deeper conclusion with application of relative coordinate system. It looks it is unnecessary here. All can be explained in absolute dimensions as well.*
As was mentioned in the paragraph 4.3. we use this part of the paper to compare our simulation to the semi-empirical fluctuating plume model. This model simulates the plume in the relative coordinate system, therefore, we do too.

*418 – 421 It is the statement that one should do the measurements close to the downwind direction and on longer street segments when going further from the source. Did it really required LES?*
No, the paragraph is saying one should know on which distances from the source the plume is expected to be most fluctuating, so if the measurements have to be performed there extra care can be taken to have a good set of plumes that will average out these fluctuations. It is a bit un-intuitive to discern this shape from the Eq. 10 but that equation says the plume is less fluctuating very close to the source, then there is a range of distances where the plume oscillates the most, and then again a range where is less fluctuating. LES was useful for confirmation of this conclusion.

*423 There are some numbers from the LES indicating the distance 300m as the less "fluctuative" area of plume – how does it refer to fig 5 where authors present substantial changes in plume direction and real measurement where only 10% of plume were captured?*
Figure 5. shows plumes from the simulation that hasn't been used in any of the analysis on plumes or their statistics. The reasons for that are explained in section 4.1.

*How the conclusion about weak meandering in a distance farther than 1500 came out. As it is very important for the topic of the article authors should give more numbers and physics here.*
We refer to the section 4.2.3. where we analyse the meandering of the plume. There it can be seen that the influence of meandering in the total plume spread either levels off (y direction) or tends to zero (z direction) (especially visible on Fig. 11).

*426 "..far from the source….highest chance of measuring" It looks good when one can fly with drone but is it referring to ground base measurements as well?*
Far from source, the plume oscillates around its center of mass, which is lifted from the ground as the plume gets mixed through the BL. So the highest chance of measuring the plume is with a drone. However, since the plume is well mixed through the BL, there is still a very good chance of measuring it at the ground.

*438 It has inverted U-shape*
We do not agree with the reviewer on this comment. If one takes a transect over y and on a single height z on one of the plumes show in in Fig.13, it is visible the middle of the plume has lower values of $i_c$ than its sides. So if the $i_c$ at a single height was plotted on a $y - i_c$ graph it would be visible it has a U shape.

*448 beta was not introduced earlier, p as well but it is explained in line 449*
Added an explanation for $\beta$ for where it is introduced.

*Fig.13 The X distances are 100,600 and 3000m? Is the ic Marro in distance of 100m indeed 600m wide? I doubt eq 10 gives this shape, even unoptimized. Scales are again making the mess from plume shapes. How does the asymmetry of (icr,LES) in Y dimension come at large distance?*

$i_{c,Marro}$ is a concentration fluctuation intensity model develop for point sources. Marros function does not have a solution on distances close to the source. The way the shape functions ($\zeta_y$ and $\zeta_z$) were defined in their paper (Marro et al. 2015, https://doi.org/10.1007/s10546-015-0041-9), close to the source they have zero values which kills off most of the terms in eq. 10 and $i_{cr}$ has the same value as the $i_{cr0}$ or the fluctuations at the centerline, which per their definition is zero. The $i_{crMarro}$ at 78 m is actually showing divergence, or the infinitely wide plume, not a plume 600 m wide. As previously mentioned, we do not have a true point source as we are bound by the resolution. That is why we undergo the pain of optimization of the function for a source with a physical shape.

*Fig.14 Linear scale is not working well for lognormal distributions. If gamma function is proposed as a PDF shape some statistical quantitative verification is required. Please use the test.* Fig.14 has been re-scaled to the semi-logarithmic scale and added to the text.

The mean p-value calculated using the Kolmogorov–Smirnov test amounted to 0.26 in the range $x =$ [100, 1500] m downwind from the source. The optimal range of downwind distances where the Gamma distribution is the best fit for the p.d.f has also been found in the LES study of Ardeshiri et al. (2020, https://doi.org/10.1007/s10546-020-00537-6) where they connected the start of this range with the maximum of $i_{cr}$ on the centerline. Following the results on Fig 14, the Gamma functions at the inlet height (z = 3 m) seem to reasonably fit the concentrations away from the plume centerline. However, the p-values obtained from the Kolmogorov–Smirnov test on most all downwind distances had values had values below 0.05 which indicates that the Gamma in this case is not the best fit. This discussion has been added to the text.

*Conclusions are mostly descriptive and completely not in the agreement with measurement results (discussed earlier).*

Hopefully the discussion on all the comments in this document changed the perspective on the agreement of the conclusions with the rest of the paper.

*490 "…which can help…" please use the specific arguments and show the areas where LES gives additional valuable information which cannot be acquired directly at the field.*

We found that the LES can aid in understanding on which heights the plumes centerline can be expected depending on the downwind distance and where the plume is expected to be most fluctuating which then requires a higher number of measurements to average out the atmospheric variability from the mean plume. This has been added to the text.

*497 "…correctly." There is no quantitative prove that it gives better results than simple Gaussian model.*

The performance of the simple Gaussian model was not in the scope of our study. However, we refer again to two studies in which the Gaussian model was validated against LES data Caulton et al. (2018, https://doi.org/10.5194/acp-18-15145-2018) and Raznjevic et al. (https://doi.org/10.5194/amt-2022-25). There it has been shown that the GP model agrees well with the measured plumes only if sufficiently long averaging times are applied. Furthermore, it is shown that the

Gaussian plume model does not take into the account the change in the plume centerline position which results in wrong source estimations of the emission rates.

*507 – 510 Give the uncertainty and perform the discussion*
Uncertainties from section 4.2 are copied here.

*520 L also has some uncertainty.*
$L$ is calculated using the equation 9 in section 4.2.3. and it is calculated for every 300 s of the simulation from the domain averaged values of the parameters in eq. 9. Reported value of 1360 m is an average of the values calculated every 300 s and its standard deviation is 68 m, i.e. $L = (1360 \pm 68)$ m. Added to the text.

*Final comment: The simulations presented by author assumed that the methane is a conservative tracer which behaves in a same way as the air but usually industrial release is not following this assumption. Methane as a part of natural gas coming from the oil reservoirs deep in the crust may be much wormer than the air, it can be also colder as expanded from the point leak. So, the information about the temperature of the source gas is very important when analysing large releases with close distanaces, but not present in the paper.*
In the situation of the diffuse emissions (not a strong point source) of the oil- and gas wells that we measured with the mobile van, the concentrations are still at atmospheric levels (up to 20 ppm when "standing in the plume" (20 m downwind of the well)). With this level of concentrations, the density is negligible towards turbulence, e.g. Stull (1950, http://dx.doi.org/10.1007/978-94-009-3027-8).

---

## Author Comment (AC2)

**Authors responses to review 2**

In this document, we have repeated the reviewers comments in the *italics* and our responses are in the standard font.

*The manuscript by Raznjevic et al. focuses on the application of LES model to understand point-source methane plume behavior. It's a timely and welcomed addition that enriches our understanding of LES model in real world dispersion study. The observation data is collected near an oil well during the ROMEO campaign, and the boundary conditions in the LES model is derived from ERA5 data. The idea is interesting, and it should enrich our understanding of plume behavior. Because the LES model can provide much more details than filed observation and simple Gaussian model. However, the conclusions of this manuscript are mostly descriptive, and several statements are not thoroughly supported.*

We would like to thank the reviewer on their kind words and the time they have invested in reading and evaluating our paper. We will try do address the comments they have to the best of our abilities so that we can clarify any statements that are not well supported and be more quantitative in our conclusions.

*Specific comments:*

*1 Some basic information of the observation campaign is missing. In section 2.2, there are two basic measurement instruments: TILDAS for CH4 and GILL R3 for wind speed, TILDAS was placed at the top of the vehicle. I'm not sure if the observation data were collected when the vehicle was moving, or collected when the vehicle stopped at the observing spot. If the data were collected when the vehicle was moving, the uncertainty of the observation data is very large. The authors should provide the time series or the linearly interpolated observation data (I'm not sure if figure 6 shows the original observation data, it says CH4 measured over the read adjacent to the emitting oil-well.), including the location of the vehicle.*

The TILDAS spectrometer was placed in the vehicle, with in inlet in the from at 3 m height. A GILL 3D Windmaster (sonic anemometer), was placed at a height on 180 cm, 25m from the source but free from obstructions for windspeed and wind directions measurements. The measurements took place while driving downwind of the oil well, to collect the plumes of the oil well and the N2O tracer. If needed, we can add as supporting information the recorded tract of the vehicle as it was measuring the plume.

Yes, the data shown on Fig. 6 is directly measured data, so no linear interpolation.

*And if possible, the authors should provide some basic meteorology information, such as wind, temperature, relative humidity. Most of the meteorology information was from ERA5 data, which has a spatial resolution of 31km, and the study area is in vicinity of hills, the meteorological parameters are not homogeneous.*

Windspeed and wind direction, as shown in Fig. 6 (b) and (c), were measured onsite with the anemometer. These local measurements are needed for the emission determination. Measurements took place at a sunny day without clouds. Meteorological information as temperature, pressure and relative humidity

are derived from ERA5, which is suitable spatial resolution for defining the stability class.

*2 According to the authors, the study area is in vicinity of hills. The surrounding environments, such as topography and buildings, should be provided by the authors. Because the dispersion near the surface (the observation is about 3 m above ground level) was deeply affected by the surroundings. In some LES models, the topography options are available, for example PALM. If the topography options is available in MicroHH, the surrounding environments should be added into the model before simulation. And the simulated wind data should be improved.*
Yes, the mountainous area started 5 - 10 km N-NW of the study area. The are itself was flat grassland with very little obstacles. The space between the oil well and the road on which the measurements were taken had no obstacles that could distort the plume. In our simulation there is no orography or any kind of obstacles because we were simulating dispersion on a $4.8 \times 4.8$ km area around the source that were just flat fields. However, the reviewer is right about including the orography to minimize the errors in the wind direction that we had in the first simulation (section 4.1.). For this particular case that would mean expanding the simulation domain far beyond the studied area to include the surrounding hills. This would have as a consequence a huge increase in the computing costs because the resolution would have to be kept same to what it is now to see the fine detail of the plume close to the source. Those kinds of simulations are unfeasible at the moment. We circumvented this problem by matching our simulated winds with the observed one in the second simulation since we did not expect huge orographic influence on the instantaneous wind, rather on timescales of an hour, as the hills were some distance away. The the problem with the domain size and required resolution is encountered in other models capable of performing the LES (such as PALM). The possible topographic forcings should be adapted in the boundary conditions, but here we encountered the problem of ERA5 resolution being too low to capture the local topography. We recognize this point has not been stressed enough in the motivation part of our paper, and we will add this discussion in the results section where the comparison of modeled and measured wind is being compared (section 4.1).

*3 In figure 6, the authors have provide the time series of observed wind speed and wind direction data, compared with simulated values. It seems that the observed and simulated values are irrelevant. The authors said that was possible caused by influences from the local orography. In my opinion, the comparison is unreasonable, because the surface in ideal model is flat, while it's heterogeneous in real world. And the GILL R3 is mounted only 1.8 m above ground level which is totally influenced by the surrounding environments. If there is no other wind measurement, the validation of modeled meteorological conditions is unnecessary. Because the boundary condition in the LES model is derived from ERA5 data, and the modeled profiles show very good agreement with ERA5 profiles.*
Yes, we mentioned the influence on the local orography on the wind, but since the first mountains were located 5 - 10 km N and NE from the site, we expected only the influence on longer period fluctuations. We refer to the paper of Nastrom et al. (1987, https://doi.org/10.1175/1520-0469(1987) 044<3087:AIOTEO>2.0.CO;2) where they have shown that the mountains can have influence on atmospheric variability extending 4 to 80 km. This was added to the paper in the Result section.
The nudging towards the ERA5 profiles in our simulation was only weak. Our goal was to have a well developed turbulent boundary layer with only the mean characteristics constrained in the boundary conditions. For this reason it was important to verify with the measurements that a good wind profile was delivered by the model.
As mentioned above, the area around the oil well was flat grassland. No obstacles that could influence the wind or the plume were present. The land there, as anywhere, has some roughness to it (from the

grass for example) but this is calculated into our model. The LES in MicroHH uses a surface model that is constrained to rough surfaces and turbulent flows. The model computes the surface fluxes of the horizontal momentum components and the scalars using Monin–Obukhov similarity theory.

*Technical corrections:*
*Figure1 Please add more information about the surrounding environments (such as google earth), the measurement (such as a photo of the vehicle).*
Added both the Google Earth image of the oil well and the photo of the vehicle.

*Line 93-111 The discussion of meteorological situation over the whole Europe is unnecessary. The low wind speeds during the campaign is enough for the explanation of local influence.*
It is a bit superfluous have the pressure map over the whole Europe, we agree with that. We will shorten the discussion down only to the area above Romania and at the measurement site.

*Line 111 The contribution of the large scale advection was . . .*
Fixed.

*Line 121 What does B20 flasks means*
B20 flasks are 20L cylinders; volume at 100bar = 2000L gas. This was added to the text.

*Figure 2 Since you have an sonic anemometer (GILL R3), the observed sensible heat flux can be calculated.*
Yes, this is true, we agree with the reviewer. We have added to here a figure showing the comparison of surface sensible heat flux from the ERA5 data and the 10 min averages that were available from the sonic anemometer. Both measured and fluxes from ERA5 show very similar values for the duration of measurements. Since the fluxes from the measurements were available for only fraction of the day, and we have used Fig. 2 to describe the meteorological conditions in the simulation, we have chosen to show only ERA5 values.

[Figure]

Figure 1: Sensible heat flux from ERA5 (hourly values) and calculated from the sonic data (10 min averages) at the location of the measurements.

*Line 172 Since the study area is heterogeneous (hills), the roughness length should be much larger.*
We hope this is clarified with the discussion above. But, again, the studied area was flat grassland. These roughness lengths correspond with that kind of terrain.

*Line 250 Replace "Them" with "The".*
Replaced.

*Line 251-258 It can not be seen that the wind angle in the idealized simulation fluctuates comparable to the observations.*
Standard deviations in wind direction are $\sigma_{WD,meas}$ = 16.9° for the measured wind and for the LES $\sigma_{WD,LES}$ = 18.6°, so they are quite similar. This has been added to the text.

*Line 315 The boundary layer height (hBL) is determined by what ? The profile of potential temperature ? The profile of sensible heat flux ?*
The boundary layer height has been calculated as a the maximum of domain- and time- averaged vertical profile of potential temperature gradient. This quantity is easily obtained from the MicroHH output. The explanation is added to the text.

*Line 542 LES can not reproduce meteorological conditions. The meteorological conditions are boundary conditions.*
LES has periodic boundary conditions on all variables, apart from the scalar that represents the plume. This means that all the momentum, heat and humidity that exits through the left boundary enters back into the simulation through the right. So in that sense there are no true boundary conditions in this simulation. The meteorological conditions were imposed by nudging the simulation towards them so the simulation does not drift too much from them. This is why we point out that LES was successful in reproducing the real meteorological conditions.